# CQM: Curriculum Reinforcement Learning with a Quantized World Model

**Seungjae Lee, Daesol Cho, Jonghae Park, H. Jin Kim**
Seoul National University
Automation and Systems Research Institute (ASRI)
Artificial Intelligence Institute of Seoul National University (AIIS)
{ysz0301, dscho1234, bdfire1234, hjinkim}@snu.ac.kr

## Abstract

Recent curriculum Reinforcement Learning (RL) has shown notable progress in solving complex tasks by proposing sequences of surrogate tasks. However, the previous approaches often face challenges when they generate curriculum goals in a high-dimensional space. Thus, they usually rely on manually specified goal spaces. To alleviate this limitation and improve the scalability of the curriculum, we propose a novel curriculum method that automatically defines the semantic goal space which contains vital information for the curriculum process, and suggests curriculum goals over it. To define the semantic goal space, our method discretizes continuous observations via vector quantized-variational autoencoders (VQ-VAE) and restores the temporal relations between the discretized observations by a graph. Concurrently, ours suggests uncertainty and temporal distance-aware curriculum goals that converges to the final goals over the automatically composed goal space. We demonstrate that the proposed method allows efficient explorations in an uninformed environment with raw goal examples only. Also, ours outperforms the state-of-the-art curriculum RL methods on data efficiency and performance, in various goal-reaching tasks even with ego-centric visual inputs.

## 1 Introduction

Goal-conditioned Reinforcement Learning (RL) has been successfully applied to a wide range of decision-making problems allowing RL agents to achieve diverse control tasks [42, 1, 30]. However, training the RL agent to achieve desired final goals without any prior domain knowledge is challenging, especially when the desired behaviors can hardly be observed. In those situations, humans typically adopt alternative ways to learn the final goals by gradually mastering intermediate sub-tasks. Inspired by the way humans learn, recent RL studies [29, 10, 6] have solved uninformed exploration tasks by suggesting which goals the agent needs to practice. In this sense of generating curriculum goals, previous approaches proposed various ideas to involve providing intermediate-level tasks [10, 38], quantifying the uncertainty of observations [4, 31, 33, 25, 18], or proposing contextual distance to gradually move away from the initial distribution [15, 6].

However, previous curriculum RL studies are mostly not scalable. Namely, they suffer from serious data inefficiency when they generate curriculum goals in high dimensions. Because of this limitation, they usually rely on the assumption that manually specified goal spaces (e.g., global X-Y coordinates) and clear mappings from high-dimensional observations to the low-dimensional goal spaces are available. Such an assumption requires prior knowledge about observations and the tasks, which remains a crucial unsolved issue that restricts the applicability of previous studies.

In order to design a general curriculum solution without the need for prior knowledge about the observations, defining its own goal space for the curriculum could be an effective scheme. To do so,

37th Conference on Neural Information Processing Systems (NeurIPS 2023).

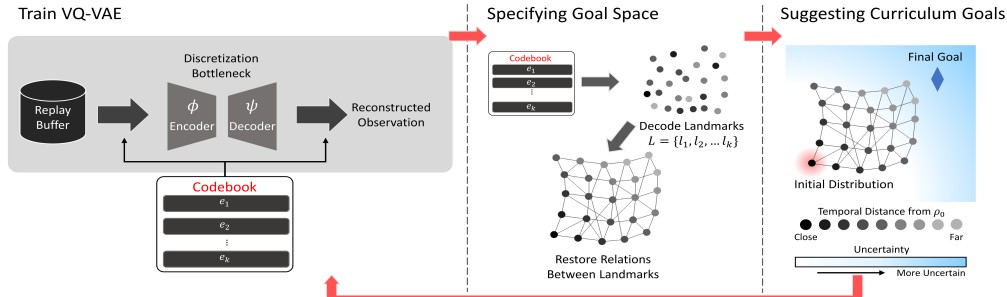

Figure 1: CQM simultaneously tackles the interrelated problems of specifying the goal space and suggesting which goal the agent needs to practice. CQM trains a VQ-VAE to form a discretized goal space and constructs a graph over it, capturing the relations between the discretized observations (landmarks). Concurrently, CQM suggests the agent which goal to practice based on uncertainty and temporal distance.

the two following operations need to be executed concurrently. (1) composing the *semantic goal space* which contains vital information for the curriculum process from the arbitrary observation space, and (2) suggesting to the agent which goal to practice over the goal space. Let us consider an agent that tries to explore an uninformed environment with final goal images only. To succeed, the agent needs to specify the semantic goal space from the high-dimensional observation space, and suggest the curriculum goals (e.g., the frontier of the explored area) over the composed goal space to search the uninformed environment. However, most previous studies focused solely on one of these, specifying the low-dimensional goal space without considering how to provide intermediate levels of goals [17, 27], or just suggesting curriculum goals in manually specified semantic goal spaces [10, 21, 6].

The challenge of simultaneously managing (1) specifying the goal space and (2) suggesting curriculum goals is that they are intimately connected to each other. If the agent constructs an ill-formed goal space from the observation space, it would be difficult to propose curriculum goals over it. Conversely, if the method fails to suggest goals to enable the agents to explore the unseen area, it would also be difficult to automatically learn the goal space that covers the uninformed environment based on the accumulated observations. Therefore, it is essential to develop an algorithm that addresses both defining goal space and providing curriculum goals concurrently.

In this paper, we propose a novel curriculum reinforcement learning (RL) method which can provide a general solution for a final goal-directed curriculum without the need for prior knowledge about the environments, observations, and goal spaces. First, our method defines its own semantic goal space by quantizing the encoded observations space through a discretization bottleneck and restoring the temporal relations between discrete goals via a graph. Second, to suggest calibrated guidance towards unexplored areas and the final goals, ours proposes uncertainty and temporal distance-aware curriculum goals that converge to the final goal examples.

The key contributions of our work (CQM: **C**urriculum RL with **Q**uantized World **M**odel) are:

- CQM solves general exploration tasks with the desired examples only, by simultaneously addressing the specification of a goal space and suggestion of curriculum goals (Figure 3).
- CQM is the *first* curriculum RL approach that can propose calibrated curriculums toward final goals from high-dimensional observations, to the best of our knowledge.
- CQM is the *only* curriculum RL method that demonstrates reliable performance despite an increase in the problem dimension, among the 10 methods that we experimented with. (Even state-based → vision-based)
- Ours significantly outperforms the state-of-the-art curriculum RL methods on various goal reaching tasks in the absence of a manually specified goal space.

## 2   Related Works

**Curriculum Goal Generation.**   Although various prior studies [41, 19, 48, 8, 45, 22] have been proposed to solve exploration problems, enabling efficient searching in uninformed environments still

remains a challenge. An effective way to succeed in such tasks with hardly observed final goals is **identifying uncertain areas** and instructing an agent to achieve the goals in these areas. To identify the uncertain areas and provide the goals sampled from them, previous studies employ uncertainty-based curriculum guidance by the state visitation counts [4, 31], absolute reward difference [37], and prediction model [32, 5]. Other approaches propose to utilize disagreements of ensembles[33, 50, 28] or sample the tasks with high TD errors [18] to generate goals in uncertain areas. An alternative way for solving the exploration tasks is to execute a **final goal-directed exploration** to propose tailored guidance. To this end, some studies perform successful example-based approaches [25, 6] or propose to minimize the distance between the final goals and curriculum goals [38, 21], measuring it by the Euclidean distance metric. Some studies also employ contextual distance-based metrics to perform final goal-directed **exploration away from the initial distribution** [15, 6].

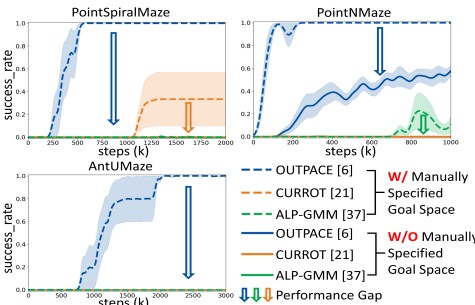

However, these methods usually assume that agents have prior knowledge about the observations and unrestricted access to manually specified semantic goal space (e.g. global X-Y coordinates) because they are not scalable to handle the high-dimensional goal spaces. For example, the meta-learning classifier-based uncertainty metrics [25, 6] suffer from distinguishing uncertain areas as the dimension of the goal space increases. Also, some of the methods rely on Euclidean distance metric [38, 21] over the goal space. Moreover, generating curriculum goals [10], employing various prediction models [32, 5, 33, 50], fitting Gaussian mixture models [37], and utilizing disagreements of ensembles-based methods [33, 50] also face difficulty in solving increasingly complex

Figure 2: When the curriculum methods are not scalable to handle the high-dimensional goal; the performance drop in the absence of manually specified goal space.

problems in high-dimensional goal spaces. Although there have been attempts to propose a curriculum in high-dimensional observations [36, 13] or include an encoder in their model-based agent [28, 14], unfortunately, these approaches do not incorporate a convergence mechanism to the final goals, which are crucial for efficient curriculum progresses.

Our method incorporates the benefits of the aforementioned methods without manually specified goal spaces: **exploring uncertain areas** and **moving away from the initial distribution** while **converging to the desired outcome**. Although there is a study that has also incorporated these three aspects [6], it retains its performance only in manually specified goal spaces, as other curriculum methods (Figure 2). (We included conceptual comparisons between CQM and more related works in Table 1 in Appendix B.)

**Discretizing Goal Space for RL.** Vector quantized-variational autoencoder (VQ-VAE) is an autoencoder that learns a discretized representation using a learnable codebook. The use of this discretization technique for learning discrete representations in RL is a recent research topic [17, 27] and has shown an improved sample efficiency. Islam et al. [17] proposes to apply VQ-VAE as a discretization bottleneck in a goal-conditioned RL framework and demonstrates the efficiency of representing the continuous observation spaces into discretized goal space. Also, Mazzaglia et al. [27] utilizes VQ-VAE to discover skills in model-based RL by maximizing the mutual information between skills and trajectories of model states.

Unfortunately, the aforementioned methods require a pre-collected dataset or extra exploration policy, which are not necessary in CQM. Training VQ-VAE with a pre-collected dataset implies that the agent has access to the full information about the task or that it already possesses an agent capable of performing the task well. Although it is possible to obtain a pre-collected dataset through a random rollout policy, this is only in the case where exploring the environments is easy enough to succeed with only random actions.

## 3 Preliminaries

We consider a Markov Decision Process (MDP) which can be represented as a tuple $(\mathcal{O}, \mathcal{A}, \mathcal{T}, \mathcal{R}, \rho_0, \gamma)$, where $\mathcal{O}$ is an observation space, $\mathcal{A}$ is an action space, $\mathcal{T}(o_{t+1}|o_t, a_t)$ is a transition function, $\rho_0$ is an initial distribution, and $\gamma$ is a discount factor. Note that the MDP above

does *not* contain a goal space, since we do not assume that the manually specified goal space is provided. Instead, we consider a discrete low-dimensional discrete goal space $\mathcal{G}$, which is defined by the agent automatically. Also, we assume that the final goal examples $o^f$ are provided by the environment, and we denote the projection of these examples into the goal space $\mathcal{G}$ as $g^f$. We represent curriculum goal as $g^c$, which is sampled from the goal space $\mathcal{G}$, and the reward function in the tuple can be represented as $\mathcal{R} : \mathcal{O} \times \mathcal{A} \times \mathcal{G} \to \mathbb{R}$. Furthermore, we denote actor network as $\pi : \mathcal{O} \times \mathcal{G} \to \mathcal{A}$, and critic network as $Q : \mathcal{O} \times \mathcal{A} \times \mathcal{G} \to \mathbb{R}$. (Thus, $Q(o, a, g)$ indicates the goal-conditioned state-action value where goal $g$, action $a$, and observation $o$ throughout this paper.)

## 4 Method

In order to provide a general solution for efficient curriculum learning, our method defines its own goal space and suggests to the agent which goal to practice over the goal space simultaneously. To compose a *semantic goal space* which reduces the complexity of observation space while preserving vital information for the curriculum process, we first quantize the continuous observation space using a discretization bottleneck (section 4.1) and restore temporal relations in discretized world model via graph (section 4.2). Over the automatically specified semantic goal space, we generate a curriculum goal and guide the agent toward achieving it (section 4.3).

### 4.1 Specifying Goal Space via VQ-VAE

In order to define a discrete low-dimensional goal space which allows a scalable curriculum with high-dimensional observations, we utilize VQ-VAE as a discretization bottleneck [46, 40, 47] as recently been proposed [17, 27]. VQ-VAE utilizes a codebook composed of $k$ trainable embedding vectors (codes) $e_i \in R^D, i \in 1, 2, ...k$, combined with nearest neighbor search to learn discrete representations. The quantization process of an observation $o_t$ starts with passing $o_t$ through the encoder $\phi$. The resulting encoded vector $z_e = \phi(o_t)$ is then mapped to an embedding vector in the codebook by the nearest neighbor look-up as

$$z_q = e_c, \quad \text{where } c = \text{argmin}_j ||z_e - e_j||_2. \tag{1}$$

The discretized vector $z_q$ is then reconstructed into $\hat{o}_t = \psi(z_q)$ by passing through the decoder $\psi$. We closely follow [17] and train quantizer, encoder, and decoder using a vector quantization loss with a simple reconstruction loss. The first term in Eq. 2 represents the reconstruction loss, while the second term represents the VQ objective that moves the embedding vector $e$ towards the encoder's output $z_e$. We update the embedding vectors $e$ using a moving average instead of a direct gradient update [17, 27]. The last term is the commitment loss, and we use the same $\lambda_{\text{commit}}$ $(= 0.25)$ across all experiments.

$$L_{\text{VQ}} = ||\psi(\phi(o_t)) - o_t||_2^2 + ||\text{SG}[z_e] - e||_2^2 + \lambda_{\text{commit}} ||z_e - \text{SG}[e]||_2^2, \quad (\text{SG : stop gradient}) \tag{2}$$

By utilizing VQ-VAE, the RL agent can specify a quantized goal space that consists of discrete landmarks $L = \{l_1, l_2, \cdots l_m\}$ in two ways. The first approach is to obtain each landmark by decoding each code as $l_j = \psi(e_j)$. Alternatively, one can obtain the landmarks by passing (encoding, quantizing to the closest embedding vector, and decoding) the continuous observations sampled from the replay buffer through the VQ-VAE. We utilize the first approach as the default, and provide the ablation study to examine the effectiveness of the second approach.

It should be noted that this set of landmarks $L = \{l_1, l_2, \cdots l_m\}$ only represents discretized observations and does *not* involve relations among the observations. We describe our approach that can better restore the temporal relations between landmarks in the next section.

### 4.2 Graph Construction over Quantized Goal Space

In this section, we present the graph construction technique over the quantized goal space to allow the agent to capture the temporal information between the landmarks of the quantized world model. We consider a graph $\mathbf{G} = (\mathbf{V}, \mathbf{E})$ over the goal space where the vertices $\mathbf{V}$ represent the landmarks

$L = \{l_1, l_2, \cdots l_m\}$ obtained from decoding discrete codes of VQ-VAE, and the edges $\mathbf{E}$ represent the temporal distance. We utilize Q-value to reconstruct the edge costs, following the method proposed in [49, 24]. If an agent receives a reward of 0 when reaching a goal and -1 otherwise, the timesteps required to travel between landmarks can be estimated using Q-value as (derivation: Appendix A)

$$\text{TemporalDist}(l_i \to l_j) = log_\gamma(1 + (1 - \gamma)Q(l_i, a, l_j)). \tag{3}$$

Using Eq. 3, we connect vertices with the distance below the cutoff threshold, and the resulting graph restores the temporal relations between the landmarks over a discretized goal space based on the temporal distance. In this way, the agent can calculate geodesic distances between landmarks,

$$\text{TemporalDist}^{\mathbf{G}}(l_0 \to l_f) = \Sigma_{(l_i \to l_j) \in \text{shortest path}(l_0 \to l_f)}\text{TemporalDist}(l_i \to l_j), \tag{4}$$

which enables better prediction of temporal distances in environments with arbitrary geometric structures. Also, to incorporate extended supports of the explored area into the graph by creating landmarks in newly explored areas, we periodically reconstructed the graph following [24].

### 4.3 Uncertainty and Temporal Distance-Aware Curriculum Goal Generation

In the previous sections, we proposed a method for specifying a discretized goal space with semantic information. It is important to provide curriculum goals located in the frontier part of the explored area to expand the graph effectively toward the final goal in an uninformed environment. To achieve this objective, we propose an uncertainty and temporal distance-aware curriculum goal generation method.

To obtain a curriculum goal from graph $\mathbf{G} = (\mathbf{V}, \mathbf{E})$ over the specified goal space, our method samples the landmarks that are considered uncertain and temporally distant from the initial distribution $\rho_0$. Thanks to the quantized world model, quantifying uncertainty in a countable goal space is straightforward and computationally light. We quantify the count-based uncertainty of each landmark as $\eta_{\text{ucert}}(l_i) = 1/(\mu(l_i) + \epsilon)$, based on the empirical distribution $\mu(l_i)$ derived from the recent observations as

$$\mu(l_i) = \frac{N(l_i)}{\sum_{i=1}^{k} N(l_i)}, \tag{5}$$

where $N(l_i)$ indicates the number of occurrences of landmark $l_i$ in recent episodes and is periodically re-initialized when the graph is reconstructed with a new set of landmarks.

Finally, we deliver the sampled landmarks as curriculum goals to the agent, considering both temporal distance and uncertainty aspects:

$$\text{argmax}_{l_i \in L^{\text{top-k}}}\left[\eta_{\text{ucert}}(l_i) \cdot u_i\right] \tag{6}$$

where $u_i$ is a uniform random variable between 0 and 1 used to perform weighted sampling, and $L^{\text{top-k}}$ represents a subset of $L$ that includes the top-k elements with the largest $\text{TemporalDist}^{\mathbf{G}}$ (Eq. 4) values from initial state distribution. Our method, based on the uncertainty and temporal distance-aware objective (Eq. 6), is capable of providing calibrated curriculum guidance to the agent even in environments with arbitrary geometric structures, without requiring prior knowledge of the environment or a manually specified goal space. Furthermore, the curriculum guidance makes composing the goal space easier, illuminating the unexplored areas and vice versa.

**Convergence to the final goal.** The curriculum objective in Eq.6 provides a calibrated curriculum towards unexplored areas. In addition to this frontier-directed method, providing final goal-directed guidance can further improve the efficiency of exploration especially when the agent sufficiently explored the environment, i.e., the supports of the final goal and explored area start to overlap [35, 6]. In order to acquire the capability to propose a final goal-directed curriculum, we gradually shift the direction of exploration from the frontier of the explored area to the final goal distribution. To do so,

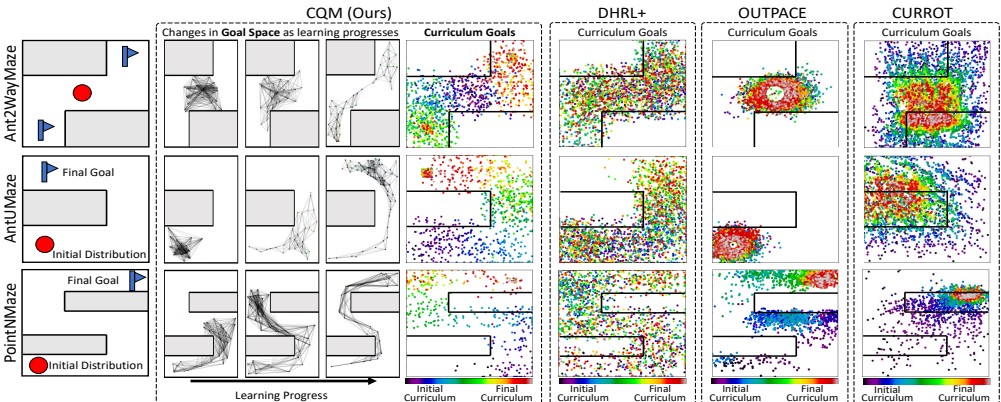

Figure 3: Left: changes in the discretized goal space of the CQM(ours) as learning progresses. Right: visualization of the curriculum goals proposed by the CQM and baseline algorithms.

we determine whether to provide the *curriculum goal* $g^c \in \mathcal{G}$ that is sampled via Eq. 6 or the *final goal* $g^f = \phi(o^f) \in \mathcal{G}$ which the environment originally provided ($\psi(g^c) \in \mathcal{O}, o^f \in \mathcal{O}$).

We utilize a mixture distribution of curriculum goals, following the approach proposed in [35],

$$p_{g^{c\prime}} = \alpha p_{g^f} + (1 - \alpha) p_{g^c}, \tag{7}$$

where $p_{g^f}$ is the distribution of the final goal, and $p_{g^c}$ is the distribution of curriculum goals. The mixture ratio $\alpha$ measures whether the achieved goal distribution $p_{ag}$ "covers" the final goal distribution using KL divergence as $\alpha = 1/\max\left(\beta + \kappa D_{\mathrm{KL}}(p_{g^f}\|p_{ag}), 1\right)$. When the support of achieved goal distribution $p_{ag}$ (= visited state distribution) covers that of the final goal distribution $p_{g^f}$, $\alpha$ produces a value close to 1, and a value close to 0 when the supports of both distributions are not connected.

By combining the curriculum goal objective (Eq. 6) with the mixture strategy (Eq. 7), our approach generates instructive curriculum goals towards unexplored areas and provides the curriculum goals $g^{c\prime}$ to the agent that "cover" the final goal distribution at the appropriate time when the agent is capable of achieving the final goal.

**Planning over the graph**   As presented above, CQM constructs a graph to restore the temporal relations between landmarks (Section 4.2) and utilizes it to calculate geodesic distances (Eq. 4). In addition to these benefits, we highlight that the graph can also provide the strength of planning, which allows the agent to reason over long horizons effectively [9, 16, 13, 49, 3, 24].

To generate a sequence of waypoints for achieving each goal, we perform shortest path planning (Dijkstra's Algorithm), following the details proposed in the previous graph-guided RL method [24]. Consider a task of reaching a curriculum goal $g^c \in \mathcal{G}$ from the current observation $o_0 \in \mathcal{O}$. CQM first adds the encoded observation $\phi(o_0)$ to the existing graph structure. Then, it finds the shortest path between the curriculum goal and current observation to return a sequence of waypoints $(\phi(o_0), w_1, ..., w_n, g^c)$ where $n$ indicates the number of waypoints in the shortest path. Finally, the agent is guided to achieve each decoded waypoint $\psi(w_i)$ during $\mathrm{TemporalDist}(\psi(w_{i-1}) \to \psi(w_i))$ (Eq. 3) timesteps, rather than achieving the curriculum goal directly. In other words, the RL agent produces goal-conditioned action $\pi(\cdot|o_t, w_i)$, where $o_t$ and $w_i$ is observation and the waypoint (acting as a goal) respectively. After reaching the final waypoint $\psi(w_n)$, the agent receives the original curriculum goal, $g^c$. The only change when the agent attempts to achieve the final goal $g^f$ is that $g^f$ comes at the end of the sequences, $(\phi(o_0), w_1, ..., w_n, g^f)$, rather than $g^c$. In this way, the proposed approach not only provides a tailored curriculum for achieving the final goal but also allows the agent to access more elaborate instructions (waypoints) for practicing each curriculum goal.

## 5   Experiments

The main goal of the experiments is to demonstrate the capability of the proposed method (CQM) to suggest a well-calibrated curriculum and lead to more sample-efficient learning, composing the goal space from the arbitrary observation space. To this end, we provide both qualitative and quantitative

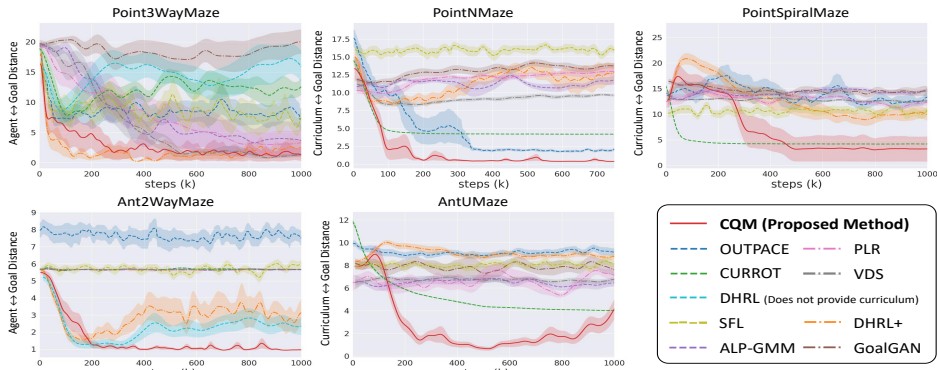

Figure 4: (**Lower is better**) Distance from the curriculum goals to the final goals (PointNMaze, PointSpiralMaze, and AntUMaze). In the 'n-way' environments with multiple goals, we provide $l2$ distance between the agent and the final goal at the end of the episodes, since calculating the average distance from the curriculum goal to multiple final goals is not possible.

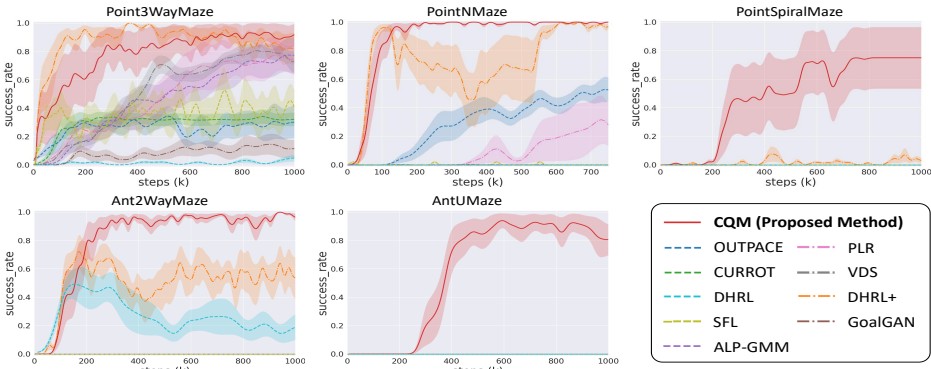

Figure 5: (**Higher is better**) Success rates of the results. The curves of baselines are not visible in some environments as they overlap each other at zero success rate. Shading indicates a standard deviation across 4 seeds.

results in seven goal-reaching tasks including two visual control tasks, which receive the raw pixel observations from bird's-eye and ego-centric views, respectively. (refer to Appendix C for the detailed configurations of each task.)

We compare our approach with previous curriculum RL methods and previous graph-guided RL methods. We do *not* provide manually specified goal space in any of the environments; the agent could not map its global X-Y coordinates from the full observation which includes all the state variables for the RL agents (e.g. angle and angular velocity of the joint, position, velocity ...). Also, the results of CQM and the baselines that utilize external reward functions (all the methods except OUTPACE [6]) are obtained by using sparse reward functions. For the baselines that could not be applied in vision-based environments [24, 6], we utilize an extra autoencoder with auxiliary time-contrastive loss [44, 13].

The baselines are summarized below: **OUTPACE** [6] proposes uncertainty and temporal distance-aware curriculum learning based on the Wasserstein distance and uncertainty classifier. **CURROT** [21] interpolates the distribution of the curriculum goals and the final goals based on the performance of the RL agent. **GoalGAN** [10] proposes the goals with appropriate levels of difficulty for the agent Using a Generative Adversarial Network. **PLR** [18] selectively samples curriculum goals by prioritizing the goals with high TD estimation errors. **ALP-GMM** [37] selects the goals based on the difference of cumulative episodic reward between the newest and oldest tasks using Gaussian mixture models. **VDS** [50] proposes the goals that maximize the epistemic uncertainty of the action value function of the policy. **DHRL** [24] constructs a graph between both levels of HRL, and proposes frontier goals when the random goals are easy to achieve. However, the original DHRL could not generate curriculum goals without the help of the environment. Thus we evaluated a variant of DHRL (DHRL+) with a modified frontier goal proposal module and architecture (Appendix D.3), in

addition to the original DHRL. **SFL** [13] constructs a graph based on successor features and proposes uncertainty-based curriculum goals. (refer to Appendix D for detailed implementations)

## 5.1 Experimental Results

First, we visualize the quantitative results to show whether the proposed method successfully and simultaneously addresses the two key challenges: 1) specifying goal space from arbitrary observation space and 2) suggesting a well-calibrated curriculum to achieve the final goal. Figure 3 illustrates the curriculum goals and changes in discrete goal space (graph) of CQM as learning progresses. Each node in the graph consists of the decoded embedding vectors of VQ-VAE, and each edge represents reachability between the decoded embeddings. The graphs of CQM in the figure gradually expand towards unexplored areas as the learning progresses, since the calibrated curriculum goal induces the agent to explore the unexplored area. In the opposite direction as well, the capability of providing proper curriculum goals on arbitrary geometric structures is facilitated by a compact goal space that contains semantic information which enables estimating the uncertainty and temporal distance well. As a result, our method provides tailored curriculum guidance across the environments, while the baselines suffer from the absence of the manually specified goal space.

We also provide the quantitative results in Figures 4 and 5. Figure 4 indicates that the proposed method (CQM) can suggest a tailored sequence of goals that gradually converges to the final goal distributions while instructing the agent to achieve the increasingly difficult goals. Also, as shown in Figure 5, ours consistently outperforms both the prior curriculum method and graph-RL methods. It is noticeable that CQM is the only method that shows robust performance to the variation of the goal dimension, while other methods suffer from serious data inefficiency, especially in the tasks with higher-dimensional goal space (suffering more in Ant (29dims) compared to Point (6dims)).

**Curriclum learning and planning in visual control tasks.** To validate the performance of the RL agent and the quality of generated curriculum goals in higher dimensional tasks, We conducted two additional vision-based goal-reaching tasks. PointNMaze-Viz receives only ego-centric view images to reach the goal, while PointSpiralMaze-Viz receives bird's-eye view images. Figure 6 visualizes the curriculum goals in the order of the episodes, and how the agent utilizes the benefit of planning over the discrete goal space in order to achieve the curriculum goals. To achieve an

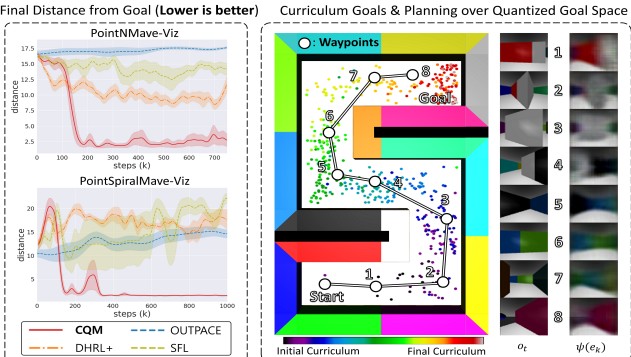

Figure 6: Left: the distance from the agent to the final goals (**Lower is better**). Right: visualization of curriculum goals and waypoints of planning over the graph (CQM).

image-based final goal (**Goal: 8**), the agent generates the sequence of images (**{1, 2, 3, ..., 8}**) as waypoints, and tries to achieve the waypoints sequentially.

Interestingly, despite a significant increase in the observation dimension, CQM does not suffer from significant performance degradation in terms of data efficiency, which indicates that CQM effectively reduces the complexity of goal space by constructing a semantic goal space. We emphasize that the performance of our algorithm does not show significant differences between state-based and image-based environments (Compare PointNMaze in Figures 4 and 6). Another interesting point is that CQM can fully enjoy the advantage of planning over the discretized goal space, even in vision-based control tasks where the agent does not receive information about its global X-Y coordinates explicitly. These results validate that CQM possesses robust performance in terms of the dimensionality of the goal space, and the capability in extracting temporal relations between the discretized landmarks.

## 5.2 Ablation Studies

**Curriculum guidance.** First of all, we examine how important curriculum guidance is for an agent to solve goal-conditioned tasks. As shown in Figure 7, when only the final goal is provided without a tailored curriculum (**-w/o Curriculum**), the RL agent has difficulty achieving the final goal directly.

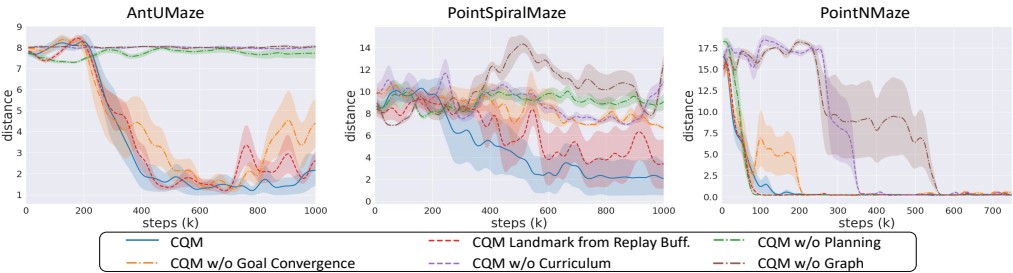

Figure 7: **(Lower is better)** Ablation study: the distance from the agent to the final goals at the end of the episodes.

Furthermore, we found that providing curriculum guidance greatly affects the goal space specification module and the absence of a curriculum leads to the ill-formed discrete goal space that barely covers only the observations near the initial distribution. We provide these qualitative results in Figures 13, 14 (Appendix E).

**Types of the discrete goal sampling method.** The proposed method (CQM) can use two approaches to sample the landmark to form the discrete goal space as introduced in Section 4.1. The first approach is to decode the embedding vectors of the codebook $l_{1:m} = \psi(e_{1:m})$, and the other approach is to sample an observation batch from the replay buffer and pass it through VQ-VAE to quantize it (**-Landmark from Replay Buff.**). As shown in Figure 7, there is no significant difference between them in terms of data efficiency. However, in terms of the stability of learning, utilizing the decoded embeddings of VQ-VAE shows better performance in some environments.

**Effect of the goal convergence method.** To provide a final goal-directed exploration in addition to the naïve curriculum toward the frontier areas, CQM includes a goal convergence module that guides the agent to practice the final goal after the agent sufficiently explored the environment (Section 4.3). Based on the KL divergence between the achieved goal distribution and the final goal distribution, CQM calculates the ratio of the mixture between the final goals and the frontier goals (the ratio of providing final goals as learning progresses is presented in Figure 11 in Appendix E). As shown in Figure 7, the absence of the final goal convergence method (**-w/o Goal Convergence**) results in unstable performance, since the agent repeatedly practices unexplored areas instead of converging towards the final goal even after the explored area "covers" the final goal distribution.

**Effect of Graph Construction and Planning.** Finally, we examine the effect of constructing graphs and planning on the performance of RL agents. As explained in section 4.2, CQM not only utilizes the decoded embedding vectors from VQ-VAE as a set of discretized observations but also forms a graph by capturing the temporal relations between the discrete observations. First, we evaluated CQM without graph (**-w/o Graph**), which does not construct a graph and measure the distance between the landmarks through naïve temporal distance prediction based on Q values (TemporalDist), rather than the geodesic distance over the graph (TemporalDist$^{\mathbf{G}}$). Also, we evaluate CQM without planning (**-w/o Planning**) since ours can optionally utilize the benefit of planning and reason over long horizons using the graph. As shown in Figure 7, CQM shows better performance than both CQM without a graph and CQM without planning, especially in some long-horizon tasks (AntUMaze and PointSpiralMaze).

## 6 Conclusions

To solve the complex control tasks without the need for a manually designed semantic goal space, we propose to solve both issues of specifying the goal space and suggesting the curriculum goals to the agent. By constructing the quantized world model using the decoded embedding vectors of the discretization bottleneck and restoring the relations between these, CQM considers both the uncertainty and temporal distance and has the capability of suggesting calibrated curriculum goals to the agent. The experiments show that the proposed method significantly improves performance on various vision-based goal-reaching tasks as well as state-based tasks, preventing the performance drop in the absence of a manually specified goal space.

**Limitations and future works.** While CQM shows great potential in addressing the limitations of previous studies, more research could further develop it. One area that could be explored is the use of reward-free curriculum learning methods, since CQM still requires minimal human efforts such as defining a success threshold to train agents. Also, this study only used single-code representations with VQ-VAE which would possess a limited capacity of representations, so expanding CQM to include multiple-code representations with discrete factorial representations could be an interesting future direction.

## 7    Acknowledgement

This work was supported by Korea Research Institute for defense Technology Planning and advancement (KRIT) Grant funded by Defense Acquisition Program Administration(DAPA) (No. KRIT-CT-23-003, Development of AI researchers based on deep reinforcement learning and establishment of virtual combat experiment environment)

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

# A Algorithm and Derivation

## A.1 How CQM Works?

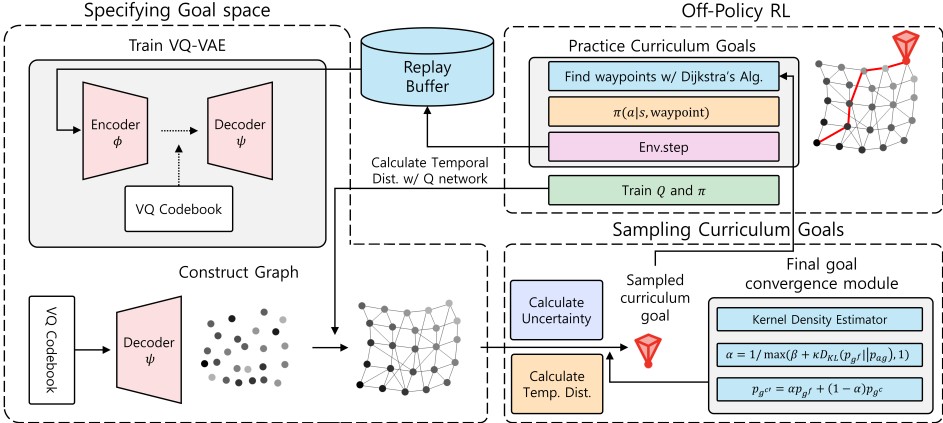

Figure 8: The overall diagram of CQM

### A.1.1 How is data collected?

1. CQM starts with empty replay buffers.
2. After performing a step in the environment, the observed transition is stored in the replay buffer. (line 18 in Algorithm 1.)

### A.1.2 How is the goal space learned?

1. Sample batch from replay buffer. (Line 26 in Algorithm 1.)
2. Train VQ-VAE with the batch via Eq. 2. (Line 27 in Algorithm 1.)

### A.1.3 How is the graph constructed?

1. Decode each vector embedding in VQ-Dictionary (the decoded embeddings are the landmarks).
2. Using Eq. 3, connect the landmarks with the distance below the cutoff threshold.

### A.1.4 How does the agent get the curriculum goal?

1. Calculate the uncertainty of each node (landmarks) in the graph (by Eq. 5).
2. Calculate the distance of the landmarks from the initial area (by Eq. 4).
3. Sample a curriculum goal which is considered temporally distant and uncertain, among the nodes (landmarks) in the graph (by Eq. 6).
4. Based on the $\alpha$ value from A.1.6, the decision is made whether to provide the agent with a curriculum goal or a final goal. Then provide the selected goal to the agent.

### A.1.5 How does the agent utilize the graph?

1. After the agent obtains the curriculum goal, it can start exploring the environment.
2. CQM first finds a sequence of waypoints to achieve the curriculum goal (utilizing Dijkstra's algorithm).
3. The agent is guided to achieve each waypoint, and finally, tries to achieve the curriculum goal.

### A.1.6 How does the curriculum goal converge to the final (desired) goal?

1. At the beginning of the learning, we have some samples of the final goals (e.g. the picture taken at the end of the maze.)

2. To get $p_{g^f}$, fit a kernel density estimator (KDE) to estimate the distribution of the final goal.

3. To get $p_{ag}$, fit a kernel density estimator (KDE) to estimate the distribution of the explored area.

4. Calculate KL divergence, and then, calculate $\alpha$ in Eq. 7.

5. Utilize $\alpha$ when we get the curriculum goal (d)

## A.2 Algorithm

---
**Algorithm 1** Overview of CQM
---

1: **Input:** final goal examples $g_f \in p_g^f$, RL replay buffer $\mathcal{B}$, VQ-VAE replay buffer $\mathcal{B}_{\text{VQ}}$, actor network $\pi$, critic network $Q$, embeddings of VQ-VAE $e_{i:m}$, total trainig episodes $N$, encoder $\phi$, decoder $\psi$, Environment horizon $H$, graph update cycle $M$

2: **for** iterations $= 1 \cdots N$ **do**

3:     sample curriculum goal $g^c$ from landmarks $l_{1:m} = \psi(e_{1:m})$ using Eq. 6

4:     `Env.reset()`

5:     **if** `random.uniform(low = 0, high = 1)` $< \alpha$ **then**

6:         $g \leftarrow g^f$

7:     **else**

8:         $g \leftarrow g^c +$ random noise

9:     **end if**

10:    get waypoints $(\phi(s_0), w_1, ..., w_n, g) \leftarrow$ `Dijkstra'salgorithm`$(s_0, g)$

11:    **for** $t = 0 \cdots H - 1$ **do**

12:       **if** achieved current waypoint $w_i$ or tried more than $\text{TemporalDist}(\psi(w_{i-1}) \rightarrow \psi(w_i))$ to achieve $w_i$ **then**

13:          current waypoint index $i \mathrel{+}= 1$

14:       **end if**

15:       $a_t \leftarrow \pi(\cdot|s_t, w_i)$

16:       `Env.step`$(a_t)$

17:    **end for**

18:    $\mathcal{B} \leftarrow \mathcal{B} \cup \{s_0, a_0, s_1...\}$, $\mathcal{B}_{\text{VQ}} \leftarrow \mathcal{B}_{\text{VQ}} \cup \{s_0, s_1...\}$

19:    **if** $N\%M == 0$ **then**

20:       update $\alpha$ using kernel density estimator (KDE) [39] from Scikit-learn [34] with Gaussian kernel following [35]

21:       Update Graph $\mathbf{G}(\mathbf{V}, \mathbf{E})$, where the vertices are the landmarks $l_{1:m} = \psi(e_{1:m})$, and the costs of the edges are calculated from Eq. 3 (connect each edge if the distance is below the cutoff threshold)

22:    **end if**

23:    **for** $i$=0,1,...,P **do**

24:       Sample a minibatch b from $\mathcal{B}$

25:       Train $\pi$ and $Q$ with b

26:       Sample a minibatch $\text{b}_{\text{VQ}}$ from $\mathcal{B}_{\text{VQ}}$

27:       Train VQ-VAE (encoder $\phi$ and decoder $\psi$) with $\text{b}_{\text{VQ}}$

28:    **end for**

29: **end for**

---

## A.3 Derivation

**Derivation of Eq. 3.** Let $Q$ be a state-action value function, $l_{1:m}$ be landmarks and $a \in \mathcal{A}$ be an action that an agent executes. If the policy requires $n$ steps to reach $l_j$ from $l_i$, the state-action value with discount factor $\gamma$ can be represented as

$$Q(l_i, a, l_j) = (-1) + (-1)\gamma + (-1)\gamma^2 + \cdots + (-1)\gamma^{n-1} = -\frac{1 - \gamma^n}{1 - \gamma}. \tag{8}$$

Thus, we can recover the temporal distance between $l_j$ and $l_i$ (= $N$ steps) as $\gamma^n - 1 = (1 - \gamma)Q(l_i, a, l_j)$, and we finally get

$$\text{TemporalDist}(l_i \to l_j) = log_\gamma(1 + (1 - \gamma)Q(l_i, a, l_j)). \tag{9}$$

# B   Related Works

Table 1: Summarized conceptual comparisons between CQM and the previous works.

| | Specify Goal Space | Uncert. -Aware | T-Dist -Aware | Final Goal -Drect. Curriculum | Curriculum Proposal | Reason over Long Lorizon |
|---|---|---|---|---|---|---|
| GoalGAN[10] | ✗ | ✗ | ✗ | ✗ | GAN | ✗ |
| CURROT[21] | ✗ | ✗ | ✗ | ✓ | Uniform | ✗ |
| PLR[18] | ✗ | ✗ | ✗ | ✗ | Buffer | ✗ |
| VDS[50] | ✗ | ✓ | ✗ | ✗ | Buffer | ✗ |
| ALP-GMM[37] | ✗ | ✗ | ✗ | ✗ | GMM | ✗ |
| HGG[38] | ✗ | ✗ | ✗ | ✓ | Buffer | ✗ |
| SkewFit[36] | ✗ | ✓ | ✗ | ✗ | VAE | ✗ |
| SFL[13] | ✗ | ✓ | ✗ | ✗ | Graph | ✓ |
| DHRL[24] | ✗ | ✗ | ✓ | ✗ | Graph | ✓ |
| L3P[49] | ✓ | ✗ | ✗ | ✗ | ✗ | ✓ |
| DGRL[17] | ✓ | ✗ | ✗ | ✗ | ✗ | possible (+HRAC) |
| Choreographer [27] | ✓ | ✗ | ✗ | ✗ | ✗ | ✓ |
| OUTPACE[6] | ✗ | ✓ | ✓ | ✓ | Buffer | ✗ |
| **CQM (ours)** | ✓ | ✓ | ✓ | ✓ | Graph | ✓ |

Table 1 compares our approach and previous curriculum goal generation methods [10, 21, 18, 50, 37, 38, 36, 6], graph-guided RL methods [13, 49, 24], and representation learning methods using VQ-VAE [17, 27]. The characteristics compared include:

- Specify Semantic Goal Space: whether the method can extract a compact goal space from the high-dimensional goal space.
- Ucert-Aware: whether the curriculum considers uncertainty.
- T-Dist-Aware: whether the curriculum considers temporal distance.
- Final Goal-Directed Curriculum: Whether the method utilizes target curriculum distribution for curriculum goal convergence.
- Curriculum Proposal: The source of the proposed curriculum goals.
- Reason over Long Horizon: Whether the method can reason over long horizons.

# C   Environment Details

- PointNMaze: The agent receives the observation consisting of the angle, angular velocity, XY position, and XY velocity of the agent. The agent is initialized in [0, 0] and the final goal is located at [0, 8]. The agent achieves the final goal when it comes within 0.5 from the final goal. Also, the size of the map is 12 by 12. The maximum timestep per episode is 150.
- PointSpiralMaze: It shares the other aspects with PointNMaze, but the final goal is located in [8, 16] and the size of the map is 12 by 20. The maximum timestep per episode is 250.
- Point3WayMaze: It shares the other aspects with PointNMaze, but the final goal is sampled randomly among [24, -8], [24, 0], and [24, 8]. The map has multiple dead ends and the size of the map is 32 by 24. The maximum timestep per episode is 200.
- AntUMaze: Agent receives the observation consisting of joint angle, joint angular velocity, XYZ position, and XYZ velocity of the agent. The agent is initialized in [0, 0] and the final goal is located at [0, 8]. The agent achieves the final goal when it comes within 1.0 from the final goal. Also, the size of the map is 12 by 12. The maximum timestep per episode is 350.
- Ant2WayMaze: It shares the other aspects with AntUMaze, but the final goal is sampled randomly among [4, 4] and [-4, -4]. The map has multiple dead ends and the size of the map is 20 by 12. The maximum timestep per episode is 200.

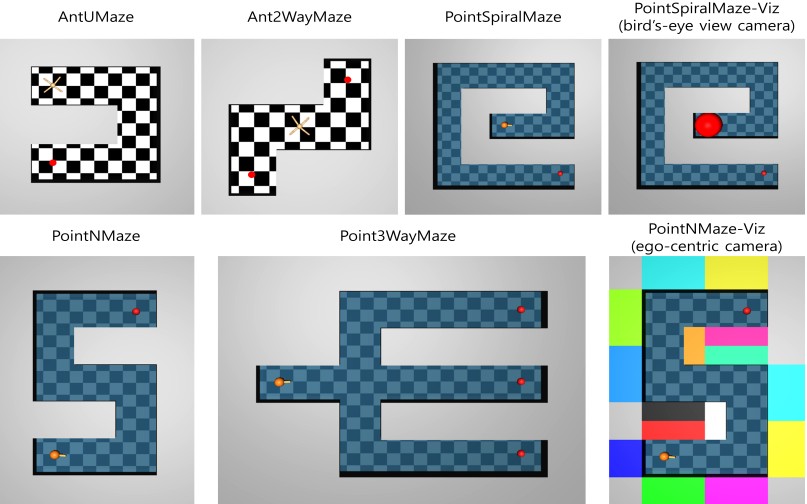

Figure 9: Top-down view of each environment used for evaluation. 'n-way' environments have multiple goals, and the final goal of an episode is generated randomly among them.

- PointNMaze-Viz: The agent receives the ego-centric camera input. The resolution of the input is 64 x 64. This environment is modified from PointNMaze, by coloring the walls. The maximum timestep per episode is 200.

- PointSpiralMaze-Viz: The agent receives the Top-down view camera input. The resolution of the input is 64 x 64. This environment is modified from PointSpiralMaze, by converting the actions of the agent from the polar coordinates to global XY coordinates and increasing the visual size of the agent. The maximum timestep per episode is 250.

## D  Implementation Details

### D.1  Learning VQ-VAE

**Replay buffer for VQ-VAE**    To train the VQ-VAE for quantizing continuous observations, we employ a separated (smaller size) replay buffer which contains recent trajectories compared to the original replay buffer for the RL agent. By employing the separated replay buffer, we can provide VQ-VAE with more recent observations, resulting in a better reflection of the areas recently explored by the RL agent.

**Code resampling for VQ-VAE**    When we use decoded embedding vectors as landmarks to discretize the goal space for CQM, VQ-VAE could ignore some of the codes due to index collapse. Then, the unused codes form landmarks in physically infeasible locations and occupy the capacity of the codes unnecessarily, leading to potential inefficiencies in the model's performance. Thus, we utilize the code resampling method [20] across all the experiments to address such problems of index collapse [20] in training VQ-VAE models.

The code resampling module in CQM first keeps track of inactive codes that have not been used during the previous M rollouts. Subsequently, the inactive codes are re-initialized with the embeddings of the recent observations $\phi(o)$ with the probability of $d_q^2(\phi(o))/\Sigma_o d_q^2(\phi(o))$, where $d_q^2(\phi(o))$ indicates the Euclidean distance from the closest code $(d_q(\phi(o)) = \min_{i \in \{1,...,k\}} ||\phi(o) - e_i||_2^2)$. We refer the reader to [27] for detailed explanations of code resampling.

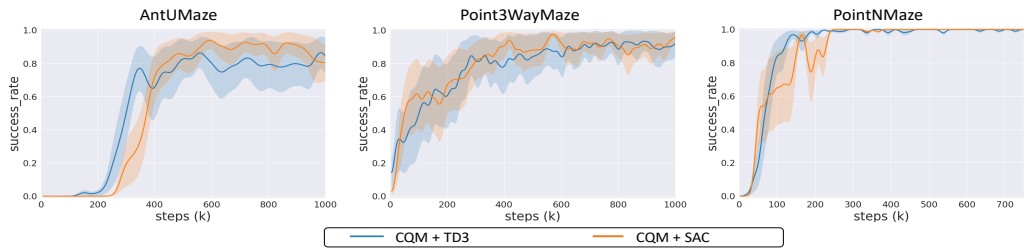

Figure 10: CQM with different RL algorithms.

## D.2 Sampling landmarks

We adopt the graph construction module of prior work [49, 24]. Inheriting their implementations, we also employ a landmark sparsification technique which is based on the Greedy Latent(Node) Sparsification algorithm [2, 49]. The detailed algorithm is shown in Algorithm 2

---

**Algorithm 2** Greedy Latent(Node) Sparsification [49, 24]

---

1: **Input:** set of states $\{e_1, e_2, ..e_k\}$, sampling number k
2: `Selected = []`, `DistList = [inf, inf, ... inf]`
3: **for** $i = 1$ **to** $k$ **do**
4:     `Farthes` $\leftarrow$ argmax(`DistList`)
5:     add `Farthest` to `Selected`
6:     `DistFromFarthest` $\leftarrow [\mathrm{TemporalDist}(\mathbf{Farthest} \rightarrow \psi(e_{1:m}))]$
7:     `DistList` = ElementwiseMin(`DistFromFarthest`, `DistList`)
8: **end for**
9: **return** `Selected`

---

## D.3 State-based Goal-Reaching Tasks

When we train CQM in state-based goal-reaching tasks, we utilize two well-established algorithms (TD3 algorithm [11] for Point- environments and SAC [12] for Ant- environments), following the baselines (TD3: [24], SAC: [21]). We note that CQM can be built on general off-policy RL algorithms [26, 12, 11] and changing the RL agents for CQM does not lead to a performance drop as shown in the figure 10.

Also, following [24], we utilize separate Q-networks for graph construction and policy learning. When the agent is not yet competent to achieve some goals, the experience of failure in the replay buffer can lead to an overestimation of the temporal distance between the landmarks. Thus, prior work utilizes separate Q-networks for graph construction and policy learning. This simple technique maintains two different Q-networks which are trained with different ratios of hindsight experience replay (HER) [1] and prevents the usage of failure trajectories to estimate the temporal distances between the landmarks which can spoil the graph construction.

**Baselines.** Since the original implementations of some algorithms [13] utilize discrete action policies, we replace them with continuous policies for comparison in the continuous control tasks. Also, for the baseline that could not generate curriculum goals without the help of the environment (DHRL) [24], we modify the frontier goal sampling module of the original baseline. Specifically, we modified the frontier goal-shifting module in the DHRL to provide a curriculum goal in every episode, not only when an easy goal is given from the environment. This modification ensures that the agent can get curriculum goals even when the environment does not provide random goals. Additionally, we empirically found that the high-level agent aggravates the performance since it suffers from providing high-dimensional subgoals. Therefore, we utilize a variant DHRL (DHRL+) with a modified frontier goal-shifting module and without a high-level. For the other baselines, we follow the official implementations (OUTPACE and CURROT) [21, 6] or the implementations from `https://github.com/psclklnk/currot` (PLR, VDS, GoalGAN, ALP-GMM).

## D.4 Vision-based Goal-Reaching Tasks

For the experiments in the vision based-based goal-reaching tasks, the agent needs to be provided with the curriculum goals corresponding to vision inputs. However, we observed that the decoded images from VQ-VAE sometimes have inferior resolution compared to the original images, which could lead to inferior performances. Thus, during the training process for the vision-based control, we utilized the observations that are mapped to each embedding as landmarks, instead of the decoded embeddings as curriculum goals. This technique does not require additional assumptions or computational costs, as the only overhead is saving the original images of each code. Although the mappings from observations to embedding vectors are many-to-one, we empirically found that there is no performance degradation even if we randomly pick one of the observations per embedding vector as a landmark.

**Baselines.** Since the official implementations of some baselines are unable to solve the vision-based goal-reaching tasks, We experimented with the additional encoder which can encode the high-dimensional observations into compact latent vectors. Considering the insights from prior representation learning research [23, 43], using a naïve autoencoder may not be fair, so we incorporated an auxiliary loss that allows for better representation learning for RL agents. To this end, we utilize an autoencoder with time-contrastive loss [7, 44] as used in SFL [13] which also performed exploration in vision-based goal-reaching tasks. Specifically, we employ the following auxiliary triplet loss to train the encoder for the baselines,

$$||\phi(o^a) - \phi(o^p)||_2^2 + m < ||\phi(o^a) - \phi(o^n)||_2^2, \tag{10}$$

where $o^a$, $o^p$, and $o^n$ represent anchor, positive pair, and negative pair respectively. Also, we utilize a margin parameter $m = 2$, following [13].

## D.5 Computational Resources

Our experiments have been performed using an NVIDIA RTX A5000 and AMD Ryzen 2950X, and the entire training process took approximately 0.5-2 days, depending on the tasks.

Table 2: Hyperparameters for CQM

| # of initial rollouts | 20 | HER [1] future step | 150 |
|---|---|---|---|
| batch size (state) | 1024 | batch size (IMG) | 128 |
| HER ratio critic Q | 0.8 | HER ratio graph Q | 1.0 |
| max graph node | 300 | graph update cycle $M$ | 5 |
| critic hidden dim | 256 | discount factor $\gamma$ | 0.99 |
| critic hidden depth | 3 | RL buffer $mathcalB$ size | 2500000 |
| actor $\phi$ learning rate | 0.0001 | critic $Q$ learning rate | 0.001 |
| interpolation factor (target Q) | 0.995 | target network update freq | 10 |
| actor update freq | 2 | # of VQ-VAE embeddings | 128 |
| VQ-VAE latent dimension | 64 (-Viz: 32) | RL optimizer | adam |

# E    Additional Experimental Results

We provide quantitative results for the ablation study in Figures 13 and 14. As we analyzed in Section 5.2, CQM without each module shows inferior results across the tasks. Without a goal convergence module, the agent often has difficulty in terms of progressing toward the desired final goals. Also, without planning, the agent often has difficulty in achieving long-horizon tasks.

As shown in Section 4.3, our algorithm includes a goal convergence module based on KL divergence between the explored area and the region corresponding to the final goal (Eq. 7) in order to perform final goal-directed exploration. Figure 11 represents the ratio of final goals given to the agent instead of curriculum goals, which is calculated according to the following equation.

$$\alpha = 1/\max\left(\beta + \kappa D_{\mathrm{KL}}(p_{g^f}||p_{ag}), 1\right) \tag{11}$$

Table 3: Task specific hyperparameters for CQM

|  | PointSpiral | PointN | PointSpiralViz | PointNViz |
|---|---|---|---|---|
| cutoff threshold for node connection | 7 | 10 | 5 | 5 |
| random noise for curriculum goal | 4 | 2 | - | - |
| VQ-VAE buffer $\mathcal{B}_{VQ}$ size | 1e5 | 1e4 | 1e5 | 1e4 |
| $\beta$ for mixture ratio $\alpha$ | -20 | -20 | -3 | -3 |
| $\kappa$ for mixture ratio $\alpha$ | 1 | 1 | 2e-3 | 2e-3 |
|  | Point3Way | AntU | Ant2Way |  |
| cutoff threshold for node connection | 10 | 30 | 30 |  |
| random noise for curriculum goal | 2 | 2 | 2 |  |
| VQ-VAE buffer $\mathcal{B}_{VQ}$ size | 5e4 | 1e4 | 1e4 |  |
| $\beta$ for mixture ratio $\alpha$ | -20 | -3 | -3 |  |
| $\kappa$ for mixture ratio $\alpha$ | 1 | 2e-3 | 2e-3 |  |

As shown in Figure 11, the ratio of providing final goals gradually increases as the learning progresses. This allows the agent to practice the final goal instead of exploring unexplored areas when the agent acquires the capability to pursue the final goals at the end of the curriculum.

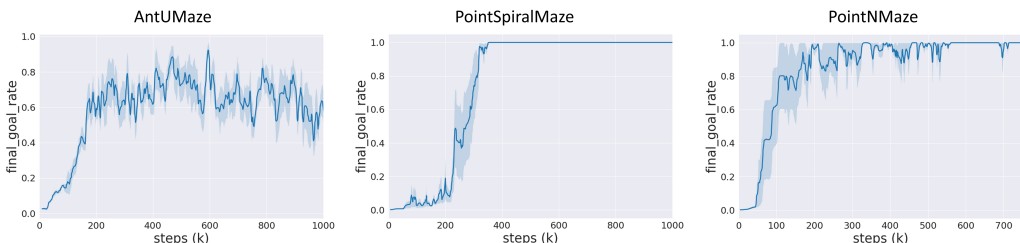

Figure 11: The ratio of providing final goals rather than curriculum goals as learning progresses (Eq. 11)

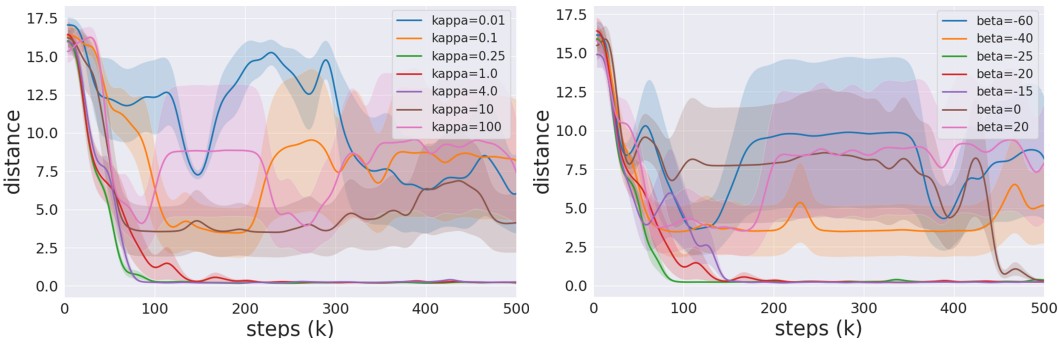

Figure 12: Ablation study: hyperparameter sensitivity analysis (PointNMaze, Lower is better)

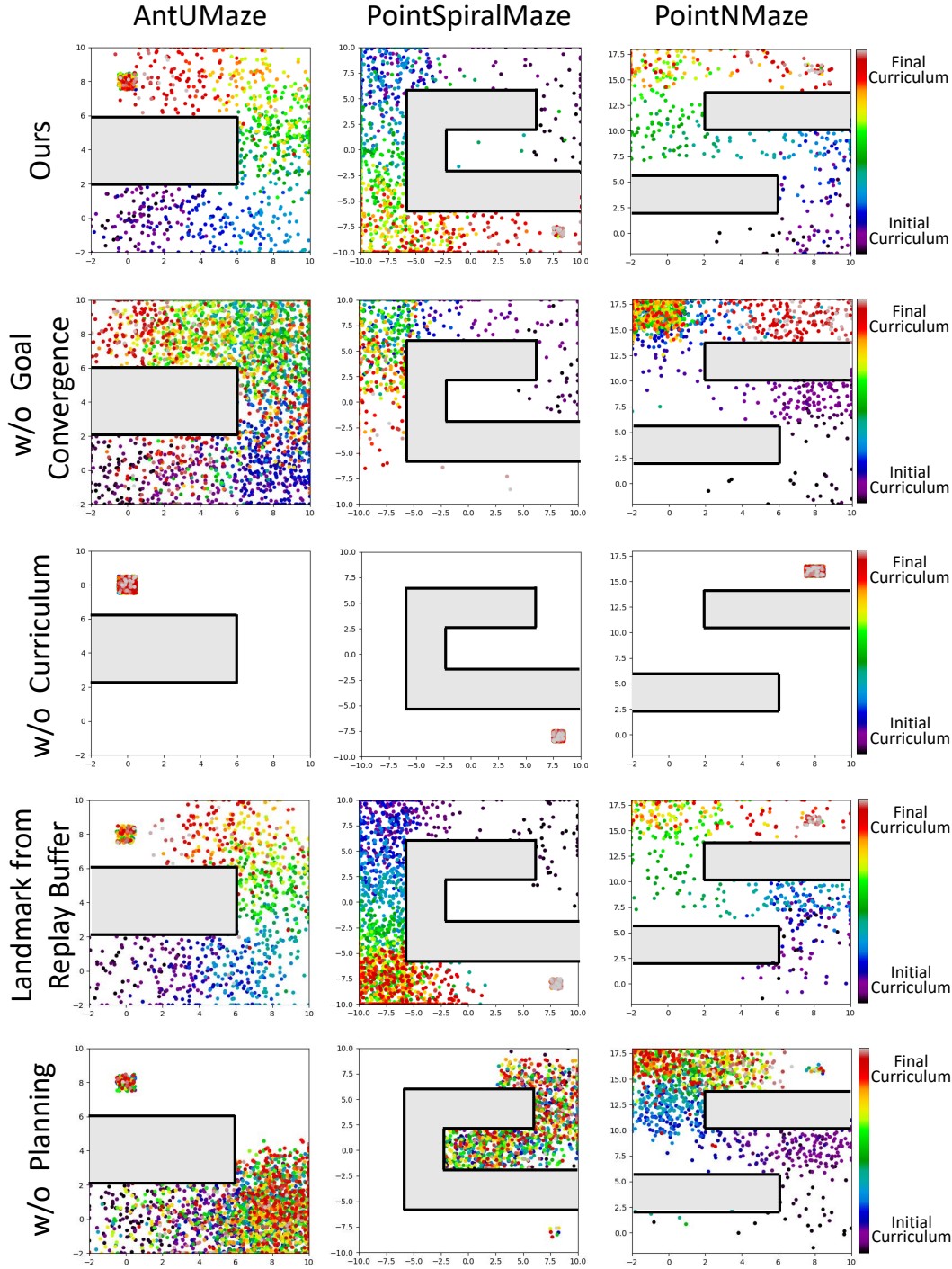

Figure 13: Ablation study: visualization of the curriculum goals proposed by the CQM.

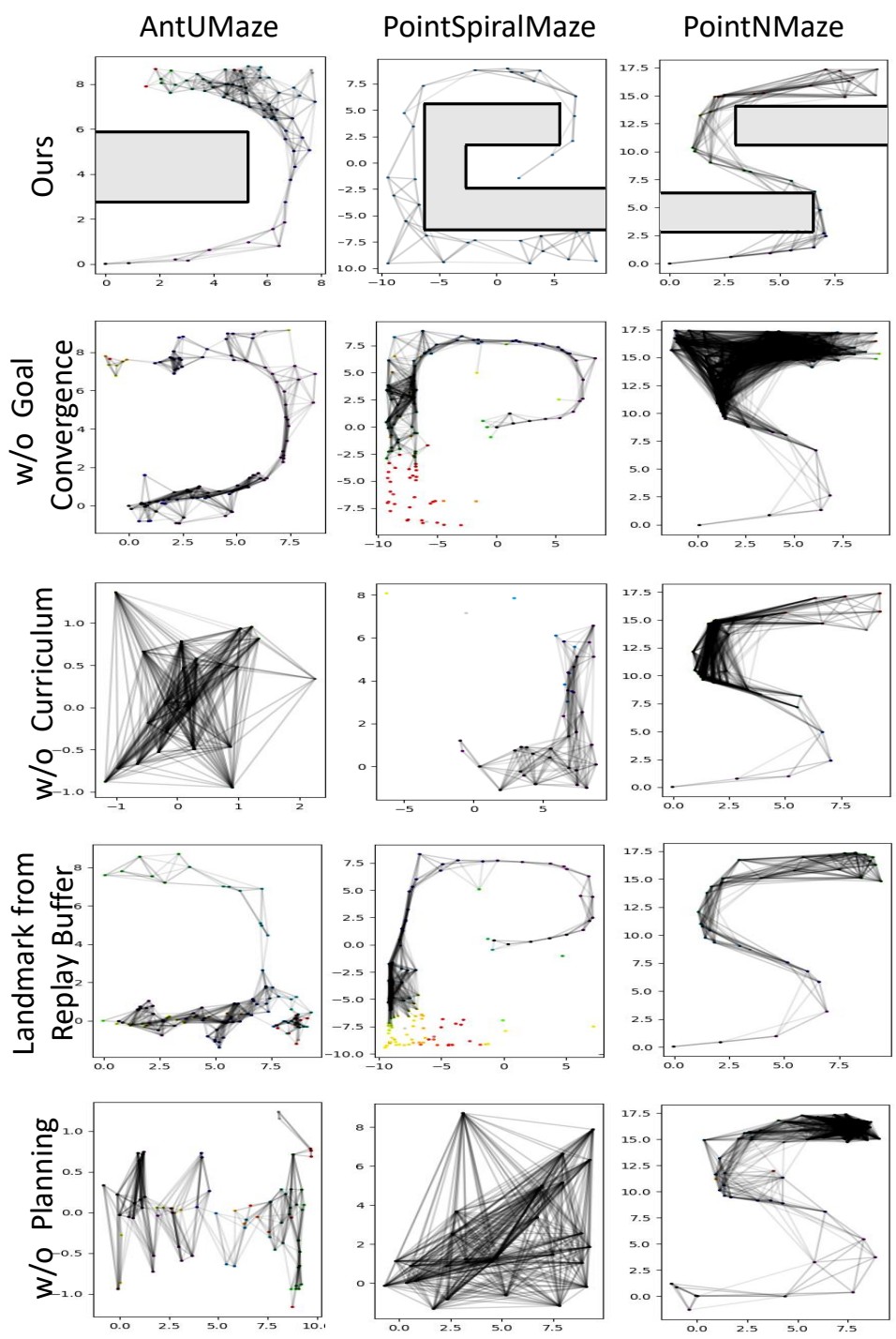

Figure 14: Ablation study: changes in the discretized goal space of the CQM(ours) as learning progresses.

