# OpenReview forum: "CQM: Curriculum Reinforcement Learning with a Quantized World Model"
_NeurIPS.cc/2023/Conference — NeurIPS 2023 poster_

### Official Review · Reviewer_vggh · 2023-07-04

**Soundness:** 3 good
**Presentation:** 3 good
**Contribution:** 3 good
**Rating:** 6
**Confidence:** 5

**Summary:**

- The proposed method addresses the challenge of learning a curriculum in goal-conditioned policies, which is a significant problem. Previous research on curriculum in goal-conditioned policies has often overlooked the importance of learning the underlying semantic goal space. This paper builds upon a recently proposed method that utilizes VQVAE to learn the semantic goal space and extends it to incorporate the learning of a curriculum. By selecting a sequence of sub-goals, this curriculum-based approach aids in achieving the final goal.

- The paper compares the proposed method to many different SOTA baselines.



**Strengths:**

- The paper is very well-written.
- The paper effectively compares the proposed method with various baselines, providing a thorough analysis.
- Additionally, the paper conducts ablations to examine the impact of different design decisions made in this research.

**Weaknesses:**

- It will be valuable to assess the effectiveness of the proposed method on both manipulation tasks and more intricate navigation tasks like 3D environments (ex. Habitat, AI2-Thor etc)
- It seems restrictive to the reviewer to use single-code representations with VQ-VAE so would be interesting to see how the results change if using multiple codes like in DGRL. (The reviewer understands this is mentioned as future work, but this seems a severe limitation if the proposed method works only with single code and not multiple codes).

- A minor comment: To enhance clarity, it is recommended to explicitly mention in the introduction that prior research, such as DGRL [1], has proposed and emphasised the significance of learning a semantic space for goal specification. It was only later in the paper that I realised vector quantisation had been proposed in previous work as an abstraction for goal-conditioned RL.

[1] Discrete Factorial Representations as an Abstraction for Goal Conditioned Reinforcement Learning, https://arxiv.org/abs/2211.00247



**Questions:**

- I'm curious about how the performance varies by varying the number of codes in VQ-Dictionary.
- It will also be useful to study if the performance of the proposed method can be improved by using self-supervised objectives for learning visual representations as done in DGRL.

**Limitations:**

Refer to Weaknesses.

---

> ### Author Rebuttal · Authors · 2023-08-09
>
> Dear reviewer vggh
>
> We sincerely appreciate your constructive and insightful comments. We found them extremely helpful. We prepared our response below:
>
> ---
>
> **Q1. It seems restrictive to the reviewer to use single-code representations with VQ-VAE so would be interesting to see how the results change if using multiple codes like in DGRL.**
>
> **A1.** Although we have left utilizing multi-code representation for CQM as future work, we agree that providing some insights into CQM's compatibility with multi-code representation would be highly valuable. To explore this possibility, we conducted experiments where we expanded the single-code representation into a multi-code representation without tuning any other hyperparameters. The numbers of embedding vectors in the VQ-dictionary (i.e. codebook) for each setting we experimented with are as follows:
>
> |  | # of embedding vectors in VQ-dictionary |
> | --- | --- |
> | 1 (single-code repr.) | 128 |
> | 2(multi-code repr.) | 32 |
> | 4(multi-code repr.) | 16 |
>
> For the experimental results, we kindly ask you to refer to *Fig. 9* in PDF uploaded to the global response. While there is some difference in convergence speed, they eventually exhibited successful performance.
>
> **Q2. I'm curious about how the performance varies by varying the number of codes in VQ-Dictionary.**
>
> **A2.** Following your comment, we attach an additional result with a varying number of embedding vectors (codes) in the VQ-dictionary. (*Fig. 10* in PDF)
>
> **Q3. It will also be useful to study if the performance of the proposed method can be improved by using self-supervised objectives for learning visual representations as done in DGRL.**
>
> **A3.** Thank you for suggesting another interesting experiment. Following your comment, we trained the encoder by adding a self-supervised objective in addition to the VQ-VAE objectives.
>
> The DGRL [3] algorithm claims to be compatible with any off-the-shelf self-supervised learning method. For this experiment, we utilized a self-supervised representation learning method (LESSON [8]) which is more recently proposed than DeepInfoMax in DGRL.
>
> $$\min_{\phi} \mathbb{E}[L_2||\phi(s_t)-\phi(s_{t+1})|| + \max(0, m-L_2||\phi(s_t)-\phi(s_{t+c})||)]$$
> ($L_2||\cdot||$ = L2 distance)
>
> LESSON adopts the technique of triplet loss: it aims to minimize the distance in the latent space between neighboring states while enforcing that the distance in the latent space between states separated by c steps (c=10 in our case) is larger than the margin parameter m.
>
> The training results (CQM+SSL) are included in the PDF (*Fig. 4*). Through the experiment, we found simply adding Self-Supervised Learning objectives could not show improved performance in the current stage. Nevertheless, we agree that exploring the appropriate SSL objectives for better performance could be a valuable research topic for future direction.
>
> **Q4. It will be valuable to assess the effectiveness of the proposed method on both manipulation tasks and more intricate navigation tasks like 3D environments (ex. Habitat, AI2-Thor etc)**
>
> **A4.** Thank you for your insightful suggestion. In the field of curriculum learning, most of the recent studies (CURROT, OUTPACE, GoalGAN, ALP-GMM, VDS…) have showcased their performance only in state-based RL settings. Thus, it's noteworthy that CQM not only outperforms these prior approaches in state-based scenarios but also demonstrates its efficacy in pixel-based environments. Thus, while we didn’t cover all the complex experiments in this study, we believe that CQM can provide valuable insights to future studies which try to establish effective curriculum learning in such complex settings.
>
> **Q5. To enhance clarity, it is recommended to explicitly mention in the introduction that prior research, such as DGRL**
>
> **A5.** We appreciate your comment. However, we would like to inform you that our manuscript already includes the references of prior studies (including DGRL [3]) in Section 2 - Related Works (lines 109-116). However, we agree that there is room for improvements (e.g. more detailed explanation of the prior works). Therefore, we will take your feedback into account and provide an improved one in the final version.
>
> ---
>
> Thank you again for your valuable and insightful review.
>
> We will gladly include these experimental findings in the final version of the manuscript, either in the main body or as an appendix.
> Also, please let us know if our responses have addressed your questions. If anything needs further clarification, please do not hesitate to let us know as soon as possible.

---

> > ### Comment · Reviewer_vggh · 2023-08-17
> > **Thank you**
> >
> > Thank you for running extra experiments. I keep my score.

---

> > > ### Author Response · Authors · 2023-08-21
> > >
> > > Thank you for replying to our response. We appreciate again your valuable suggestions and all your efforts during the review process.

---

### Official Review · Reviewer_eib8 · 2023-07-05

**Soundness:** 3 good
**Presentation:** 2 fair
**Contribution:** 3 good
**Rating:** 7
**Confidence:** 4

**Summary:**

The method works as follows. Graphs are built by (1) quantizing visual observations to create a goal space & (2) creating temporal relations over goal space vectors. Curriculum goals towards a "user"-specified goal are made using this graph.
A VQ-VAE is used to create the goal space where goals are decodings of one of k trainable embedding vectors. The authors also study an ablation where landmarks are decoded from the nearest neighbor goal encodings over replay buffer observations.
Graphs are derived by connecting edges based on a metric that exploits having rewards by -1 at all states except for the goal state. This is motivated by wanting edges to capture the geodesic between nodes.
A curriculum for exploring the graph is created by selecting nodes with high uncertainty (or low count via count-based methods) and high temporal distance. Curriculum goals are goals that maximize this sum of this uncertainty and geodesic distance away from the initial state.
The method uses Dikstra's algorithm to generate plans towards goals.

They compare against many baselines in the literature and consistently find a higher success rate. They also visualize the curriculum goals produced by their method.

**Strengths:**

originality: The method is novel to the best of my knowledge.
quality: The paper has high quality. The experiments are very though and the authors also do an analysis.
clarity: The experiments are relatively clear.
significance: The paper seems to produce better performance than many methods in the literature with similar assumptions (e.g. exploiting diikstra's algorithm).

**Weaknesses:**

The paper could strongly benefit from some figures that describe the details of the method. The method is relatively complicated (though not more complicated than comparable methods in the literature). Due its complexity, it's hard to understand how the method works from figure 1. I'm somewhat familiar with this literature so I think understand their method section was easier but other readers may find it more challenging. Another figure would strongly improve the clarity of this paper.

The baseline description method describes how the baselines work but it does not describe what comparing against each baseline (or sets of them) tells us about CQM. Without this, while the results are promising, it's hard to get a takeaway message. For example, OUTPACE also proposes an uncertainty and temporal distance aware curriculum. Comparing against OUTPACE tells us....? It might be the importance of jointly considering uncertainty and temporal distance but that isn't clear in the text. I recommend explicating this a bit more. I think it's fine if baseline methods are grouped together for describing the point of the comparison.

Given the complexity of this method, it would strongly benefit from an algorithm describing, how is data collected, how is the goal space learned, how is the graph constructed, etc. In particular, which of these are done in tandem and which are done in sequence is not clear.

Overall, this method seems promising, the motivation is clear, and the results are also promising. I think the biggest drawback is clarity right now. It's not that easy to understand the method. Right now I lean towards accept but I strongly encourage the authors to add figures/explicit algorithm/both for the methods section. This would likely push my score to a 7.

**Questions:**

Across experiments, does the agent encode first-person observations, top-down observations over a map, or both? This seems important to how you quantize the observations. If top-down, this seems like a strong assumption. Can you justify it? Methods like SFL use first-person observations. At least some of the environments that you study permit first-person observations (e.g. AntMaze). I know you do only ego-centric for PointNMave-Viz but it's not clear for the other experiments.

What assumptions does this method make over the reward function?

**Limitations:**

This limitations section is fine.

---

> ### Author Rebuttal · Authors · 2023-08-09
>
> Dear reviewer eib8
>
> We sincerely appreciate your constructive and insightful comments. We found them extremely helpful. We prepared our response below:
>
> ---
>
> **Q1. The paper could strongly benefit from some figures that describe the details of the method. Another figure would strongly improve the clarity of this paper.**
>
> **A1.** Thank you for your helpful suggestions on the need for some figures for CQM. Following your valuable comment, we have prepared a diagram (*Fig 8.* in the PDF attached to the global response) that provides a clearer understanding of each component of our algorithm. We will ensure that these visuals are incorporated into the final version of our paper. If there are any other ideas to further enhance the reader’s understanding, please do not hesitate to let us know. We will be happy to comply.
>
> **Q2. Explanation is required regarding the differences between CQM and the baselines. While the results are promising, it's hard to get a takeaway message.**
>
> We agree that a detailed comparison with the baselines can provide readers with more insightful takeaway messages. Following your suggestion, we summarized the additional comparisons below. We kindly ask you to refer to Table 1 in Appendix B (in supplementary material) when digesting the explanation below, for better understanding.
>
> **A2-1.** vs GoalGAN, PLR, ALP-GMM, and VDS
>
> Firstly, these methods each employ curriculum learning strategies, but they do not have a convergence mechanism to the final goal distribution. Thus, the agents learned with these methods repeatedly practice unexplored areas instead of converging towards the final goal even after the explored area “covers” the final goal distribution, leading to performance degradation.
>
> **A2-2.** vs CURROT: CURROT can execute curricula directed towards the final goal. However, it requires an assumption that the distance between the samples can be measured by the Euclidean distance metric. Thus, it suffers in the environments such as N-shaped or Spiral-shaped Maze.
>
> **A2-3.** vs DHRL and SFL
>
> These methods are the RL algorithms employing graphs. Unlike our method, DHRL does not autonomously generate its curriculum goals; rather, it relies on externally supplied goals that are sampled from feasible areas within the environment. Consequently, DHRL's performance heavily depends on the availability of these externally provided random goals.
>
> Although SFL allows exploration without external goal input, it does not execute final goal-directed curricula as the baselines mentioned in **A2-1.** Moreover, it solely conducts exploration based on uncertainty, unlike our approach which encompasses both uncertainty and temporal distance considerations.
>
> **A2-4.** vs OUTPACE
>
> In distinction to the studies above, OUTPACE is the only curriculum goal generation method that considers both uncertainty and temporal distance. However, there are two significant distinctions between OUTPACE and CQM:
>
> (a) OUTPACE does not have a module to learn the goal space while CQM automatically defines the goal space and quantizes it.
>
> (b) OUTPACE predicts uncertainty using the meta-learning-based method Conditional Normalized Maximum Likelihood (CNML) while CQM utilizes count-based uncertainty prediction with quantized state space.
>
> CNML struggles with substantial prediction errors when there is no prior knowledge of a manually specified goal space (since CNML is not scalable to high-dimensional observations). Conversely, our approach employs a straightforward count-based uncertainty mechanism via state space quantization, enabling the generation of suitable curriculum goals even in high-dimensional input environments without a manually specified goal space.
>
> **Q3. Clarification on each part of CQM**
>
> **A3.** To enhance reader comprehension, we have succinctly presented how each module operates in the CQM algorithm. (Due to the word limit, we kindly ask you to refer to our answer (**G-A1**) in the global response). Also, following your comment, we will incorporate this alongside the diagram mentioned in A1 in the final version.
>
> **Q4. About the environments (first-person observation / top-down observation)**
>
> **A4.** Among the two visual input environments we experimented with, PointNViz is a first-person environment, while PointSpiralViz is a third-person environment. While it is true that SFL conducted experiments in a first-person environment similar to ours, it's important to note that they used a discrete action space rather than a continuous action space like ours, making our setting a more challenging problem. (Additionally, SFL utilizes a pre-trained ResNet backbone, whereas our approach does not assume access to a pre-trained network)
>
> **Q5. What assumptions does this method make over the reward function?**
>
> **A5.** The results of CQM and the baselines in this work are obtained by using sparse reward functions. In sparse reward settings, the agent receives a reward of 0 when it succeeds in reaching a goal and -1 otherwise. The criteria for success in each environment are indicated in Appendix C. Environment Details
>
> ---
>
> Thank you again for your valuable and insightful review.
>
> We will gladly include the figures and descriptions above in the final version of the manuscript, either in the main body or as an appendix. Also, please let us know if our responses have addressed your questions. If anything needs further clarification, please do not hesitate to let us know as soon as possible.

---

> > ### Comment · Reviewer_eib8 · 2023-08-15
> > **Reviewer response**
> >
> > Thank you for your response. I am satisfied with the rebuttal. I have raised my score by 1.

---

> > > ### Author Response · Authors · 2023-08-16
> > > **Thank you for your response**
> > >
> > > Thank you for replying to our response.
> > >
> > > We are happy to hear that our response addressed your questions. We appreciate again for your valuable suggestions and all your efforts during the review process.

---

### Official Review · Reviewer_uT8t · 2023-07-06

**Soundness:** 3 good
**Presentation:** 3 good
**Contribution:** 3 good
**Rating:** 8
**Confidence:** 4

**Summary:**

This paper proposes a curriculum RL approach using a VQ-VAE to learn a goal space, and then construct a graph with the VQ-VAE codes as nodes, and a temporal distance estimate of the Q-value as weighted edges. The curriculum is then constructed by doing frontier-based exploration on this graph, by sampling goals based on their visitation count and temporal distance. In addition, the graph can be used for planning intermediate goals as waypoints for the agent. The paper is well written, and it has proper benchmarks and ablation study. The authors use the term "world model" in their paper, but I would argue they don't really train a world model, rather a representation of the observation space. I would omit the use of "world model" in a further revision of the manuscript.

**Strengths:**

The paper proposes a novel curriculum RL method, which learns a goal-space on high-dimensional pixel observations. Turning the VQ-VAE codebook into a graph, weighted by estimated temporal distance from the Q-value is a powerful idea for both curriculum learning as well as planning and long-range credit assignment.

**Weaknesses:**

- The authors use the term "world model", however what they present is a discrete latent representation of the observation space. A world model would entail to e.g. also equip it with an action-conditioned forward dynamics model, as typically used in model-based RL. I would not call this a world model.

**Questions:**

- How to get a distribution over the goal, encoding a set of exemplar goal observations and counting the bins of the VQ-VAE they get encoded in?

- In the Planning section the authors use state s_0 from S, although the previous MDP description uses observation o_0 from O. I assume these are the same or have you shifted to a POMDP setup?

- In the AntUMaze, the performance seems to deteriorate after 600k steps, i.e. the curriculum goals seem to get farther from the goal. Any insight on what is happening there?

- Since the goal space is trained as a VQ-VAE on individual observations, I think it will break when the environment is ambiguous, i.e. if the agent would have to traverse rooms, but 2 rooms would look identical, these observations would map onto the same goal and the graph might collapse?

**Limitations:**

- I think one limitation that is currently not touched upon is that this approach requires that the environment is not ambiguous (i.e. you don't see the same observations at different places in the environment).

---

> ### Author Rebuttal · Authors · 2023-08-09
>
> Dear reviewer uT8t
>
> We sincerely appreciate your constructive and insightful comments. We found them extremely helpful. We prepared our response below:
>
> ---
>
> **Q1. Regarding the term “world model”**
>
> **A1.** Thanks for your valuable comment. We understand your concern regarding the term "world model" used in our paper, which may confuse the readers as it is also used in general model-based RL.
>
> We would like to clarify that the term "world model" we used is derived from the paper "World Model as a Graph: Learning Latent Landmarks for Planning (L3P) [6]". In the L3P paper, the term "world model" refers to a graph-structured environment, since it can also be viewed as a multi-step transition at a higher-level perspective.
>
> After considering your input, we understand the need for more careful explanations when using the term "world model". We will incorporate additional explanations in the introduction of our paper, providing context and clarifying the specific meaning. If you believe that even with these clarifications, the term "world model" might still lead to confusion among readers, please kindly inform us, and we will explore alternative terms to effectively convey the intended meaning.
>
> **Q2. How to get a distribution over the goal, encoding a set of exemplar goal observations and counting the bins of the VQ-VAE they get encoded in?**
>
> **A2.** If you are asking about the goal distributions ($p_{ag}$ and $p_{g^f}$) in the section “Convergence to the final goal”(lines 201-218), we utilize kernel density estimator (KDE) and fit the KDE model to the goal distribution which is sampled from replay buffer. Also, following the prior work (MEGA [5]), we utilize the default Gaussian kernel for KDE.
>
> **Q3. In the Planning section the authors use state s_0 from S, although the previous MDP description uses observation o_0 from O. I assume these are the same or have you shifted to a POMDP setup?**
>
> **A3.** We appreciate your comment. As you mentioned, the symbols $s_0 \in S$ carry the same meaning as $o_0 \in O$. Once we have the opportunity to make manuscript revisions, we will promptly correct it.
>
> **Q4. In the AntUMaze, the performance seems to deteriorate after 600k steps, i.e. the curriculum goals seem to get farther from the goal. Any insight on what is happening there?**
>
> **A4.** Thank you for your comment. The performance drop in AntUMaze after 600k is mainly caused by the low $\alpha$ values after the discovered area covers the final goal area. This problem could be addressed with simple hyperparameter tuning, and we have just adjusted both beta and kappa to x1/2 to get better performance. We would like to share the new graph (we kindly ask you to refer to *Fig. 7.* in PDF uploaded to the global response).
>
> **Q5. Since the goal space is trained as a VQ-VAE on individual observations, it will break when the environment is ambiguous, i.e. if the agent would have to traverse rooms, but 2 rooms would look identical, these observations would map onto the same goal and the graph might collapse?**
>
> **A5.** Similar to the baselines and the prior curriculum learning studies (OUTPACE, SFL, VDS, DHRL, ALP-GMM, GoalGAN, Skew-fit, CURROT, PLR...), our algorithm also does not consider POMDP. (CQM inherits the limitations of the prior curriculum learning algorithms at this point. - We will gladly include this in the limitation part in the final version.)
>
> Consequently, in scenarios involving indistinguishable rooms, additional modules are necessary to differentiate between them. Thus, we think that exploring the curriculum learning methods for partially observable environments is also a valuable and thought-provoking research topic for future direction. We believe that incorporating methodologies that can account for past observations could be a promising direction for addressing these problems.
>
> ---
>
> Thank you again for your valuable and insightful review.
>
> Please let us know if our responses have addressed your questions. If anything needs further clarification, please do not hesitate to let us know as soon as possible.

---

> > ### Comment · Reviewer_uT8t · 2023-08-11
> >
> > I thank the authors for their detailed responses to my questions, and I appreciate the additional results they provided.

---

> > > ### Author Response · Authors · 2023-08-13
> > > **Thank you for your response**
> > >
> > > Thank you for replying to our response. We appreciate again your valuable suggestions and all your efforts during the review process.

---

### Official Review · Reviewer_paJb · 2023-07-07

**Soundness:** 3 good
**Presentation:** 3 good
**Contribution:** 2 fair
**Rating:** 5
**Confidence:** 4

**Summary:**

This paper introduces Curriculum RL with Quantized World Model (CQM), a novel approach that leverages a VQ-VAE to create a discretized goal space and constructs a graph structure over it. CQM further proposes a curriculum strategy based on uncertainty and temporal distance to guide the learning process. The authors evaluate the effectiveness of CQM through experiments conducted on variants of PointMazes and AntMazes, which serve as benchmarks for Hierarchical Reinforcement Learning.

**Strengths:**

- This paper is well-written and easy to follow up
- Extensive experiments


**Weaknesses:**

One significant weakness of this paper is the lack of clarity regarding why using the representation from VQ-VAE is suitable for graph-building. It is crucial to consider the temporal distances between nodes to adequately cover the visited state space, especially given the limited number of nodes. The proposed goal representation learning scheme by VQ-VAE does not appear to take into account these temporal distances, raising questions about the efficacy of the proposed representation in creating a meaningful semantic goal space. It would greatly benefit the paper to provide a more thorough explanation and justification for the use of VQ-VAE in graph construction and its ability to capture temporal relationships.

Furthermore, the proposed goal representation learning scheme should be compared to prior work on representation learning, such as NORL [1] or LESSON [2]. A comprehensive comparison would help establish the novelty and effectiveness of the proposed approach in relation to existing methods. Additionally, it remains unclear how the proposed goal representation learning scheme outperforms or differs from the simple approach of utilizing farthest point sampling from the replay buffer, which warrants further investigation and comparison.

Moreover, the paper introduces new hyperparameters (\alpha, \beta, and \kappa) for curriculum goal generation. However, it is unclear how these hyperparameters were determined and how the performance varies when these hyperparameters are varied. Additionally, it would be valuable to compare the costs associated with hyperparameter search between CQM and the baseline methods.

[1] Ofir Nachum, Shixiang Gu, Honglak Lee, Sergey Levine, “Near-Optimal Representation Learning for Hierarchical Reinforcement Learning”, ICLR 2019.

[2] Siyuan Li, Lulu Zheng, Jianhao Wang, Chongjie Zhang, “Learning Subgoal Representations with Slow Dynamics”, ICLR 2021



**Questions:**

- How does performance vary when \alpha and \beta are changed? Could you provide insights into the selection process for these hyperparameters? Is it possible to set them automatically?
- Regarding planning over the graph, what happens if an agent is unable to directly reach a specific node w_{i}? Could the system handle a scenario where, after TemporalDist(\phi(w_{i-1}), \phi(w_{i})), the agent conditions its behavior on reaching w_{i+1} instead?
- Could you please explain how the landmarks were sampled from the replay buffer for the "CQM Landmark from Replay Buff" depicted in Figure 7? What criteria or process were used to select the landmarks from the replay buffer?

Minor:
- It appears that there are two different symbols used for \beta, one in Equation 2 and another in \alpha = 1 / \max (\beta + \kappa D_{\text{KL}} (p_{g}^{f} || p_{ag}), 1). I would recommend using distinct symbols to avoid confusion.
- In Figure 7, the y-label representing the distance is partially hidden.


**Limitations:**

The authors adequately addressed the limitations and potential negative societal impact.

---

> ### Author Rebuttal · Authors · 2023-08-09
>
> Dear reviewer paJb
>
> We sincerely appreciate your constructive comments. We found them extremely helpful. We prepared our response below:
>
> ---
>
> **Q1. Lacks clarity on why using the representation from VQ-VAE is suitable for graph-building. VQ-VAE does not appear to take into account temporal distances.**
>
> **A1.** We appreciate your valuable comment. While we agree that explicitly incorporating temporal distance information could potentially lead to better results, we would like to kindly emphasize that our current approach with VQ-VAE already demonstrates impressive coverage of the environment and consistently outperforms the curriculum learning baselines. Also, even though the temporal distance information is not considered in learning VQ-VAE, it eventually becomes incorporated into the graph construction process when generating edges. (It is worth noting that prior studies utilizing VQ-VAE to quantize state space (DGRL[3], Choreographer[4]) have achieved favorable results without explicitly considering the temporal distance.)
>
> Furthermore, quantizing the dimensional space yields clear advantages for graph construction, notably in relation to reducing computational costs, enhancing robustness against Q-value errors, and enabling effective quantification of uncertainty. (more detailed explanations are in **A3.**)
>
> **A1-1.** We present the images of decoded embeddings demonstrating that the VQ-VAE already provides sufficient coverage of the visited state (please refer to *Fig. 3* in the PDF uploaded to the global response).
>
> **A1-2.** Also, we conducted additional experiments to incorporate temporal distance information into VQ-VAE training (*Fig. 4* in the PDF):
>
> To investigate whether temporal distance information leads to practical performance improvements, we added auxiliary self-supervised loss in learning the latent space of VQ-VAE (CQM+SSL). The auxiliary loss minimizes the distance in the latent space between neighboring observations while enforcing the distance in the latent space between temporally distant observations to become larger than the margin. We found that simply adding this objective could not show improved performance in the current stage, and we conjecture that VQ-VAE already possesses the capability to construct a graph that appropriately covers the environments well. Nevertheless, we agree that testing the effectiveness of injecting temporal distance information into CQM would be valuable.
>
> **Q2. Comparison with prior work on representation learning, such as NORL or LESSON.**
>
> **A2.** Following your suggestion, we replaced the VQ-VAE part of CQM with LESSON [8] (CQM-LESSON) and compared the performance with the original CQM. For this experiment, we modified only the representation learning part while keeping the rest of the modules intact for fair comparisons. (Since the proposed algorithm not only focused on representation learning; it encompasses curriculum goal generation and other parts). The results are provided in the PDF (*Fig. 5*). Through the experiments, we found that the original CQM consistently shows better goal-reaching performances than CQM-LESSON.
>
> **Q3. How the proposed goal representation learning scheme outperforms or differs from the simple approach of utilizing farthest point sampling (FPS) from the replay buffer.**
>
> **A3.** We compared CQM with the simple approach of utilizing farthest point sampling from the replay buffer (CQM-onlyFPS), and the results are provided in the attached PDF (*Fig. 5* and Table 1). Also, we analyzed the factors contributing to the superior performance of origin CQM compared to CQM-onlyFPS:
>
> - Since the computation complexity of FPS is $\Omega(n^2)$, where n is the number of observations in the initial sample of the FPS algorithm, using only FPS introduces significant computational overhead (Table 1.).
>
> - Additionally, FPS is not robust to temporal distance estimation error and thus, the error affects the graph construction. (This problem has also been mentioned in prior work DHRL [7].)
>
> - VQ-VAE enables count-based uncertainty measurement, but it is hard in onlyFPS option to measure the uncertainty.
>
> **Q4. Regarding the hyperparameters ($\alpha$, $\beta$, and $\kappa$)**
>
> **A4.** Our approach utilizes the mixture distribution of curriculum goals following MEGA[5] and thus, we also utilize a heuristic way to determine it. We conducted a grid search for hyperparameter tuning. *Throughout this process, we made fewer than 20 attempts per environment.* Also, the hyperparameter values found for PointNMaze were directly used without modification for all state-based Point environments, and the values for AntUMaze were directly used for all remaining environments. We provide the performance of CQM under varying hyperparameters in the attached PDF (*Fig. 6*).
>
> **Q5. What happens if an agent is unable to directly reach a specific node $w_{i}$?**
>
> **A5.** If the agent could not achieve the current waypoint $w_{i}$ even after spending TemporalDist($\psi(w_{i-1}), \psi(w_{i})$) timestep, CQM updates the current tracking waypoint to the next waypoint $w_{i+1}$ (as shown in line 12 of Algorithm 1 in Appendix A).
>
> **Q6. How the landmarks were sampled from the replay buffer for the "CQM Landmark from Replay Buff" depicted in Figure 7?**
>
> **A6**. For "CQM Landmark from Replay Buff" depicted in Figure 7 (Ablation study graph), we sampled states from the replay buffer and passed them through VQ-VAE. To do so, we randomly sampled a total of 1000 states, and after passing through VQ-VAE to quantize it. Thus, the states that fall into the same code are merged into a single state. Finally, the decoded states after this process form the nodes of the graph.
>
> ---
> Thank you again for your valuable and insightful review.
>
> Please let us know if our responses have addressed your questions. If anything needs further clarification, please do not hesitate to let us know as soon as possible.

---

> > ### Comment · Reviewer_paJb · 2023-08-14
> > **My concern has been addressed**
> >
> > Thank you for the response that addresses my concern. Please add the experimental results and discussion into the final version. I would like to raise the score.

---

> > > ### Author Response · Authors · 2023-08-15
> > > **Thank you for your response**
> > >
> > > Thank you for replying to our response. We are happy to hear that our response addressed your concern.
> > >
> > > Following your suggestion, we will discuss this in the final version. We appreciate again for your valuable suggestions and all your efforts during the review process.

---

### Official Review · Reviewer_G8ip · 2023-07-17

**Soundness:** 3 good
**Presentation:** 3 good
**Contribution:** 2 fair
**Rating:** 5
**Confidence:** 3

**Summary:**

The paper proposed a new curriculum reinforcement learning method, CQM, that uses VQ-VAE to learn a quantized goal space, constructs a graph on the quantized goals to propose curriculum goals by distance, and learns a goal-conditional policy.

**Strengths:**

- The proposed method pioneers in "auto" curriculum RL that learns the goal space and proposes goals all by itself.
- The paper is clearly written and easy to follow.
- Detailed studies on various environments are presented to demonstrate its effectiveness.
- Code is provided for reproduction.

**Weaknesses:**

- $\hat{o}_t$ in Eq. 2 should be $o_t$.
- [Vector Quantized Models for Planning](https://arxiv.org/pdf/2106.04615.pdf) has used VQ-VAE in RL. They did not explicitly generate curricula but it's very similar to the proposed method. The authors should compare and explain the difference.
- $Q(l_i, a, l_j)$ in Eq. 3 is not explained. I guess it's an expectation over the replay buffer?
- The proposed mazes are long but they mostly don't have branched dead ends, which contain unseen states but do not lead to the actual goal. [This work](https://openreview.net/pdf?id=U4r9JNyNZ7) and its experiments in more complicated mazes should be compared to.

**Questions:**

- Eq. 6 does not have a weight factor to balance the two terms. How are the two terms distributed in practice? Will one dominate the other?

**Limitations:**

The limitations have been adequately addressed.

> After rebuttal

Some of my concerns have been addressed. I appreciate the authors' efforts, but the maze in either their submission or rebuttal doesn't match the complexity of the related work I requested for comparison. Thus I'm holding my score.

---

> ### Author Rebuttal · Authors · 2023-08-09
>
> Dear reviewer G8ip
>
>
> We sincerely appreciate your constructive and insightful comments. We found them extremely helpful. We prepared our response below:
>
> ---
>
> **Q1. Differences between ‘Vector Quantized Models for Planning’ [1] and our research.**
>
> **A1.** Thank you for your helpful suggestions on the need for a comparison with the prior work. In this answer, we want to emphasize that there are very few similarities between our research (CQM) and ‘Vector Quantized Models for Planning (VQM)’ [1] except for the use of VQ-VAE.
>
> 1. Our algorithm **does NOT require a dataset to pre-train VQ-VAE before learning** an agent, while VQM needs it [1].
>     - Our algorithm enables the RL agent to start learning from an initial distribution without any prior information or dataset about the environment. To do so, we simultaneously learn VQ-VAE and the RL agent with a curriculum.
>     - On the other hand, the mentioned algorithm (VQM) [1] is divided into two stages, where in stage 1, it pre-trains VQ-VAE using a pre-existing dataset. VQM has amassed 101,325,000 episodes to gather the dataset for pre-training VQ-VAE in the Deepmind Lab Env(section 5.2.1 in the VQM paper). To do so, they randomly placed the agents in the environment and utilized A2C algorithm. *It is crucial to consider how strong of an assumption it is to possess such pre-collected data from diverse locations in the environment even before initiating the main policy.*
> 1. Ours perform goal-conditioned decision-making, while VQM [1] does not.
> 2. Ours perform curriculum learning: we generate the goals autonomously.
> 3. Our work is compatible with general off-the-shelf RL algorithms while VQM [1] is based on MuZero (which incorporates MCTS).
> 4. VQM [1] corresponds to single-layer RL, while we correspond to multiple-layer RL.
> In VQM, planning refers to determining each individual action. In contrast, planning in our algorithm carries a higher-level meaning, where nodes are connected by multiple transitions. Thus, the agents execute multiple goal-conditioned actions to facilitate movement between graph nodes in CQM. In other words, our approach can be seen as a hierarchical structure (graph at the high level, and RL agent at the low level).
>
> To the best of the author’s knowledge, ours is the first VQ-VAE work capable of exploration without relying on pre-training datasets or extra exploration policy.
>
> **Q2. This work [2] and its experiments in more complicated mazes should be compared to (e.g. branched dead ends).**
>
> **A2-1.** We appreciate your comment. However, we would like to inform you that the experiment of our work already includes branched dead ends. (e.g. Point3WayMaze, Ant2WayMaze - the maps of each environment are visualized in the Appendix in the supplementary material.)
>
> **A2-2.** Following your interesting suggestion, we created mazes (*Fig 1.* in the PDF attached to the global response) with a similar geometric shape to the maze used in the prior work [2] and evaluated our algorithm in it. Although the prior work was performed on a low-dimensional state-based environment, we took it a step further and expanded the scope to an image-based setup, thus exploring a more complex and challenging scenario.
>
> The results are shown in *Fig. 2.* in the PDF. Interestingly, even without further hyperparameter tuning from another setting (PointSpiralMaze-Viz), we achieved remarkable success in this extended setting. We believe that these results demonstrate the capability of CQM to be applied in even more intricate environments.
>
> **Q3. Explanation of $Q(l_i, a, l_j)$ in Eq. 3.**
>
> **A3.** Throughout our manuscript, the Q-function follows the notation provided below.
>
> $Q(\mathrm{current\  state}, \mathrm{action},\mathrm{goal})$
>
> Thus, $Q(l_i, a, l_j)$ indicates the goal-conditioned state-action value where goal, action, and state is $l_i$, $a$, and $l_j$, respectively. (We will incorporate this explanation into the final version.)
>
> **Q4. Questions and minor comments**
>
> **A4-1.** Regarding $\hat{o}_t$: We appreciate your comment. Once we have the opportunity to make manuscript revisions, we will promptly correct it.
>
> **A4-2.** Regarding the weight factor to balance the two terms in Eq. 6.
>
> Actually, there are various ways to implement the curriculum goal sampling part.
>
> - One approach is to implement it by adjusting the balance between the two using a weight factor. (In this case, finding a balance between the two is important.)
> - Selecting the curriculum goal based on one criterion using top-K filtering and then sampling based on the remaining criterion also yields good performance.
>
> We have found that both of these approaches can yield successful results with appropriate hyperparameters. While the equation in the manuscript corresponds to the former approach, we found that the latter approach tends to be more robust to the hyperparameters. We will include the weight factor & implementation details in the Appendix of the final version.
>
> ---
>
> Thank you again for your valuable and insightful review.
>
> Please let us know if our responses have addressed your questions. If anything needs further clarification, please do not hesitate to let us know as soon as possible.

---

> > ### Author Response · Authors · 2023-08-21
> > **Dear Reviewer G8ip**
> >
> > We hope this message finds you well.
> >
> > We appreciate the time you have taken to review our work and consider the points we raised in our rebuttal. We hope that our response has provided a more comprehensive understanding of our research and its potential contributions to the field.
> >
> > Please let us know if you need any further clarification. We appreciate again your valuable suggestions and all your efforts during the review process.
> >
> > Thank you for your consideration.
> >
> > CQM Authors.

---

### Author Rebuttal · Authors · 2023-08-09


Dear reviewers,

We sincerely appreciate your constructive and insightful comments.

We have prepared our responses at the bottom of each review you provided. This global response includes:

- [Additional Results] In this global response, **we attached a PDF containing the experimental results** for the responses to each reviewer. *(The button labeled "PDF" is at the bottom of this message)*
- [Optional] Also, In case you encounter difficulty in quickly recapping our pipeline, we have included a brief overview of the key components of our algorithm below.

Thank you again for your valuable review.
If anything needs further clarification, please let us know as soon as possible.

---
**G-A1. How CQM works?**

(a) How is data collected?

- CQM starts with empty replay buffers
- After performing a step in the environment, the obtained transition information is stored in the replay buffer. (line 18 in Algorithm 1. (Appendix A.))

(b) How is the goal space learned?

- Sample batch from replay buffer (line 26 in Algorithm 1. (Appendix A.))
- Train VQ-VAE with the batch via Eq. 2. (line 27 in Algorithm 1. (Appendix A.))

(c) How is the graph constructed?

- Decode each vector embeddings in VQ-Dictionary (the decoded embeddings are the landmarks).
- Using Eq. 3., connect the landmarks with the distance below the cutoff threshold.

(d) How the agent gets the curriculum goal

- Calculate the uncertainty of each node (landmarks) in the graph (by Eq. 5).
- Calculate the distance of the landmarks from the initial area (by Eq. 4).
- Sample a curriculum goal which is considered temporally distant and uncertain, among the nodes (landmarks) in the graph.
- Based on the $\alpha$ value from (f), the decision is made whether to provide the agent with a curriculum goal or a final goal. Then provide the selected goal to the agent.

(e) How does the agent utilize the graph?

- After the agent obtains the curriculum goal, it can start exploring the environment.
- CQM first finds a sequence of waypoints to achieve the curriculum goal (utilizing Dijkstra’s algorithm).
- The agent is guided to achieve each waypoint, and finally, tries to achieve the curriculum goal.

(f) How the curriculum goal converges to the final (desired) goal?

- At the beginning of the learning, we have some samples of the final goals (e.g. the picture taken at the end of the maze.)
- To get $P_{g^f}$, fit a kernel density estimator (KDE) to estimate the distribution of the final goal.
- To get $P_{ag}$, fit a kernel density estimator (KDE) to estimate the distribution of explored area.
- Calculate KL divergence, and then, calculate $\alpha$ in Eq. 7.
- Utilize the $\alpha$ when we get the curriculum goal (d)

**G-A2. References.**

[1] Ozair, Sherjil, et al. "Vector quantized models for planning." *international conference on machine learning*. PMLR, 2021.

[2] Kim, Seongun, Kyowoon Lee, and Jaesik Choi. "Variational Curriculum Reinforcement Learning for Unsupervised Discovery of Skills." (2023).

[3] Islam, Riashat, et al. "Discrete factorial representations as an abstraction for goal conditioned reinforcement learning." *arXiv preprint arXiv:2211.00247* (2022).

[4] Mazzaglia, Pietro, et al. "Choreographer: Learning and adapting skills in imagination." *arXiv preprint arXiv:2211.13350* (2022).

[5] Pitis, Silviu, et al. "Maximum entropy gain exploration for long horizon multi-goal reinforcement learning." *International Conference on Machine Learning*. PMLR, 2020.

[6] Zhang, Lunjun, Ge Yang, and Bradly C. Stadie. "World model as a graph: Learning latent landmarks for planning." *International Conference on Machine Learning*. PMLR, 2021.

[7] Lee, Seungjae, et al. "DHRL: A Graph-Based Approach for Long-Horizon and Sparse Hierarchical Reinforcement Learning." *Advances in Neural Information Processing Systems* 35 (2022): 13668-13678.

[8] Li, Siyuan, et al. "Learning subgoal representations with slow dynamics." *International Conference on Learning Representations*. 2020.

---

### Decision · Program_Chairs · 2023-09-21

**Decision:**

Accept (poster)

**Comment:**

After extensive discussion with the authors, all reviewers recommend acceptance and the AC agrees. Reviewers had several suggestions for additional experiments, ablations, environments, and hyperparameter analysis that the authors performed during the rebuttal period. Many of these would strengthen the manuscript and provide additional value to readers (either in the main paper or the appendix). Discussion of take-aways (e.g. the reply to eib8) and the easier-to-digest algorithm overview / diagram may also improve readability. Reviewer enthusiasm for more complex environments is likely a positive signal that CQM in image-based settings could find broader impact.